# Multi-Modal View Enhanced Large Vision Models for Long-Term Time Series Forecasting

**ChengAo Shen[1], Wenchao Yu[2], Ziming Zhao[1], Dongjin Song[3], Wei Cheng[2], Haifeng Chen[2], Jingchao Ni[1]**

[1]University of Houston, [2]NEC Laboratories America, [3]University of Connecticut
[1]{cshen9,zzhao35,jni7}@uh.edu, [2]{wyu,weicheng,haifeng}@nec-labs.com,
[3]dongjin.song@uconn.edu

## Abstract

Time series, typically represented as numerical sequences, can also be transformed into images and texts, offering multi-modal views (MMVs) of the same underlying signal. These MMVs can reveal complementary patterns and enable the use of powerful pre-trained large models, such as large vision models (LVMs), for long-term time series forecasting (LTSF). However, as we identified in this work, the state-of-the-art (SOTA) LVM-based forecaster poses an inductive bias towards "forecasting periods". To harness this bias, we propose DMMV, a novel decomposition-based multi-modal view framework that leverages trend-seasonal decomposition and a novel backcast-residual based adaptive decomposition to integrate MMVs for LTSF. Comparative evaluations against 14 SOTA models across diverse datasets show that DMMV outperforms single-view and existing multi-modal baselines, achieving the best mean squared error (MSE) on 6 out of 8 benchmark datasets. The code for this paper is available at: https://github.com/D2I-Group/dmmv.

## 1 Introduction

Long-term time series forecasting (LTSF) is vital across domains such as geoscience [1], neuroscience [3], energy [18], healthcare [28], and smart city [27]. Inspired by the success of Transformers and Large Language Models (LLMs) in the language domain, recent research has explored similar architectures for time series [40, 14, 51]. Meanwhile, Large Vision Models (LVMs) like ViT [7], BEiT [2] and MAE [12], have achieved comparable breakthroughs in the vision domain, prompting interest in their application to LTSF [4]. These approaches transform time series into image-like representations, enabling LVMs to extract embeddings for forecasting [29]. The rationale is that LVMs, pre-trained on large-scale image datasets, can transfer useful knowledge to LTSF due to a structural similarity: each image channel contains sequences of *continuous* pixel values analogous to univariate time series (UTS). This alignment suggests LVMs may be better suited to time series than LLMs, which process *discrete* tokens.

This hypothesis is partially validated by the SOTA VisionTS model [4], which applies MAE [12] to imaged time series and achieves impressive forecasting performance. This progress has spurred interests in combining imaged time series with other representations. In the past, time series have been studied through various forms: (1) raw numerical sequences [49, 30], (2) imaged representations [43, 4], and (3) verbalized (textual) descriptions [46, 10]. While they differ in modality, they represent alternative views of the same underlying data – unlike typical multi-modal data, where modalities originate from distinct sources [23]. However, these *multi-modal views (MMVs)* enable the application of large pre-trained models, such as LLMs, LVMs, and vision-language models (VLMs) [16, 33], to time series analysis, specializing them from those in conventional multi-view learning [39], where multi-view is a broader notion including both MMVs and views of the same modality (*e.g.*, augmented image views [39]). To distinguish, we use MMVs for time series throughout this paper.

39th Conference on Neural Information Processing Systems (NeurIPS 2025).

Leveraging MMVs offers two key advantages: (1) augmenting time series with alternative views can reveal patterns not evident in the original numerical data, and (2) pre-trained large models can extract complex patterns specific to certain views, such as visual representations. Motivated by these benefits and the recent success of LVMs, this work investigates the synergy of MMVs for LTSF, with a focus on incorporating LVMs. To our knowledge, integrating the visual view of time series via LVMs alongside other modalities remains underexplored. The most related effort, `Time-VLM` [52], uses a VLM (`ViLT` [16]) to encode visual view and contextual texts of time series, augmented with a `Transformer` for the numerical view. All embeddings are combined through a fusion layer. However, this simple combination strategy overlooks the unique inductive biases of individual views, leading to suboptimal performance (see §4.1). Moreover, its use of textual inputs provides only marginal improvements while introducing significant computational overhead due to the large language encoder.

We propose DMMV, a Decomposition-based Multi-Modal View Framework for LTSF, which integrates numerical and visual views in a compact architecture. We exclude the textual view due to its marginal gains in `Time-VLM` [52] and recent doubts about the effectiveness and cost-efficiency of LLMs for LTSF [36]. DMMV comprises two specialized forecasters: a numerical forecaster and a visual forecaster. The visual forecaster, inspired by `VisionTS` [4], uses `MAE` [12] – a self-supervised LVM capable of reconstructing masked images – leveraging its strong performance on continuous values (*i.e.*, pixels). Time series are transformed into images using a period-based patching technique [43], which, although effective, imposes an inductive bias on LVMs towards periodic signals. To address this, we design two DMMV variants as illustrated in Fig. 1: (a) DMMV-S (simple decomposition), which splits the time series into trend and seasonal components, assigning them to the numerical and visual forecasters, respectively; (b) DMMV-A (adaptive decomposition), which adaptively learns the decomposition via a backcast-residual mechanism aligned with the two forecasters. DMMV employs *late fusion* [17] via a gating mechanism, as intermediate fusion (*e.g.*, embedding-level) underutilizes `MAE`'s decoder, which plays a crucial role in pixel prediction. Extensive experiments show that DMMV significantly outperforms both SOTA single-view methods and `Time-VLM`, despite the latter incorporating an additional text encoder. To sum up, our contributions are as follows.

- We distinguish MMVs in time series analysis from the broader notion in conventional multi-view learning and study the emergent yet underexplored problem of MMV-based LTSF.

- We propose DMMV, a novel MMV framework that is carefully designed to harness an inductive bias we identified in SOTA LVM-based forecasters, complemented by the strength of a numerical forecaster, with two technical variants DMMV-S and DMMV-A.

- We conduct comprehensive experiments on benchmark datasets to evaluate DMMV, demonstrating its superior performance over 14 SOTA baselines and highlighting its potential as a new paradigm for MMV-based time series learning.

## 2   Related Work

To the best of our knowledge, this is the first work to explore LVMs in a decomposition-based MMV framework for LTSF. Our work relates to **LVM-based time series forecasting (TSF)**, **Multi-modal TSF**, and **Decomposition-based TSF**, which are discussed below.

**LVM-based TSF**. Various vision models, such as `ResNet` [13], `VGG-Net` [34], and `ViT` [7], have been applied to TSF [50, 47], with some studies exploring image-pretrained CNNs like `ResNet` [13], `Inception-v1` [35], and `VGG-19` [34] for LTSF [19]. The use of LVMs in this area is still emerging, with most efforts focused on time series classification (e.g., `AST` [9] uses `DeiT` [37], `ViTST` [20] uses `Swin` [26]). In contrast, LVMs have seen limited use in TSF, likely due to their lower effectiveness on low-level (*i.e.*, numerical-level) tasks. The most notable method is `VisionTS` [4], which adapts `MAE` [12] for zero-shot and few-shot TSF. Another method, `ViTime` [47], trains `ViT` [7] from scratch on synthetic imaged time series but does not explore transferring knowledge from image-pretrained LVMs. Importantly, these approaches rely solely on vision models without incorporating other views or modalities.

**Multi-modal TSF.** Recently, large VLMs such as `LLaVA` [23], `CLIP` [33], and `ViLT` [16] have been explored for time series analysis [41, 32, 56, 52]. The most relevant is `Time-VLM` [52], which builds a forecaster on `ViLT` [16] to encode numerical and visual views, along with contextual texts. While integrating rich information with a large model, `Time-VLM` demonstrates promising results in TSF.

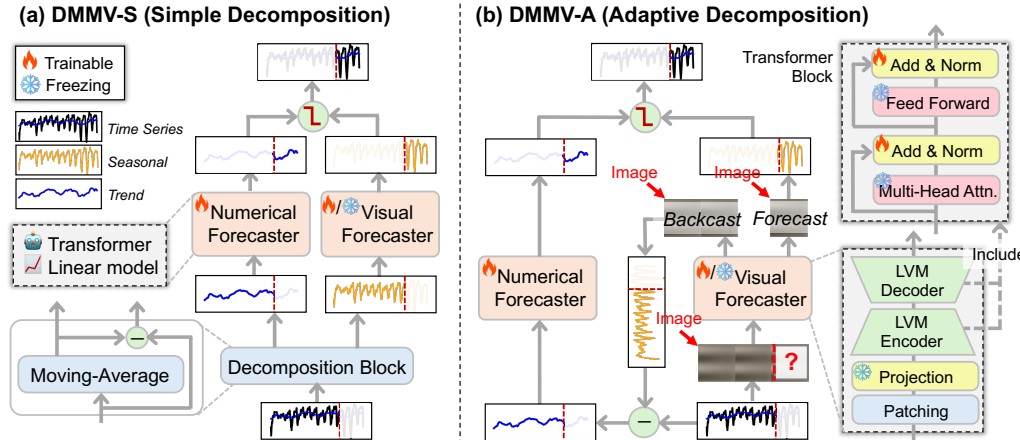

Figure 1: An overview of DMMV framework. (a) DMMV-S uses moving-average to extract trend and seasonal components. (b) DMMV-A uses a backcast-residual decomposition to automatically learn trend and seasonal components. In (b), the gray blocks are gray-scale images. "?" marks masks.

However, its fusion strategy closely follows the `ViLT` backbone and lacks time-series-specific design, leading to potentially suboptimal performance.

**Decomposition-based TSF.** Decomposition is a common technique in TSF, with seasonal-trend decomposition (STD) employed by models like `Autoformer` [44], `FEDformer` [54], and `DLinear` [49]. Recent work such as `Leddam` [48] replaces the traditional moving-average kernel in STD with a learnable one. Residual decomposition is another approach, used by `N-BEATS` [31] to reduce forecasting errors and later adopted by `DEPTS` [8] and `CycleNet` [21] for period-trend decomposition. While `SparseTSF` [22] predicts periods without explicit decomposition, `SSCNN` [6] introduces an attention-based method to extract long-term, short-term, seasonal, and spatial components. However, none of these methods incorporate LVMs. In contrast, our proposed DMMV shares insights with residual decomposition but is uniquely designed to exploit the inductive bias of LVMs for adaptive decomposition, setting it apart from prior works.

In summary, the proposed DMMV framework is distinct from existing approaches, yet integrates the strengths of pre-trained LVMs, the MMV framework, and decomposition techniques.

## 3 Decomposition-Based Multi-Modal View (DMMV) Framework

**Problem Statement**. Given a multivariate time series (MTS) $\mathbf{X} = [\mathbf{x}^1, ..., \mathbf{x}^D]^\top \in \mathbb{R}^{D \times T}$ within a *look-back window* of length $T$, where $\mathbf{x}^i \in \mathbb{R}^T$ ($1 \leq i \leq D$) is a UTS of the $i$-th variate, the goal of LTSF is to estimate the most likely values of the MTS at future $H$ time steps, *i.e.*, $\hat{\mathbf{Y}} \in \mathbb{R}^{D \times H}$, such that the difference between the estimation and the ground truth $\mathbf{Y} = \mathbf{X}_{T+1:T+H} \in \mathbb{R}^{D \times H}$ is minimized in terms of mean squared error (MSE), *i.e.*, $\frac{1}{D \cdot H} \sum_{i=1}^D \sum_{t=1}^H \|\hat{\mathbf{Y}}_{it} - \mathbf{Y}_{it}\|_2^2$.

**Preliminaries**. Masked autoencoder (`MAE`) [12] is pre-trained self-supervisedly by reconstructing masked image patches using ImageNet dataset [5]. To adapt it to LTSF, `VisionTS` [4] adopts a period-based imaging technique introduced by `TimesNet` [43]. Specifically, each length-$T$ UTS $\mathbf{x}^i$ is segmented into $\lfloor T/P \rfloor$ subsequences of length $P$, where $P$ is set to be the period of $\mathbf{x}^i$, which can be obtained using Fast Fourier Transform (FFT) on $\mathbf{x}^i$ [43] or from prior knowledge on sampling frequency. The

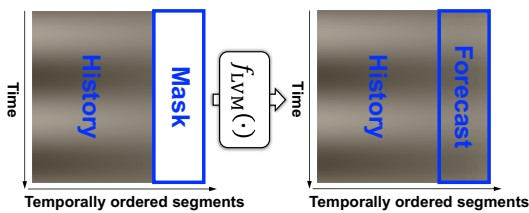

Figure 2: An illustration of an LVM forecaster

subsequences are stacked to form a 2D image $\mathbf{I}^i \in \mathbb{R}^{P \times \lfloor T/P \rfloor}$. After standard-deviation normalization, $\mathbf{I}^i$ is duplicated 3 times to form a gray image of size $P \times \lfloor T/P \rfloor \times 3$, followed by a bilinear interpolation to resize it to an image $\tilde{\mathbf{I}}^i$ of size $224 \times 224 \times 3$ to fit the input requirement of `MAE`. As Fig.

2 shows, the forecast is achieved by reconstructing a right-appended masked area of $\tilde{\mathbf{I}}^i$, corresponding to the future horizon of $\mathbf{x}^i$. The forecast $\hat{\mathbf{y}}^i \in \mathbb{R}^H$ can be recovered from the reconstructed area by de-normalization and reverse transformation. The forecast of MTS $\mathbf{X}$ is achieved by forecasting over $\mathbf{x}^1, ..., \mathbf{x}^D$ in parallel, following the channel-independence assumption [30].

**An Inductive Bias.** Due to period-based imaging and the spatial consistency enforced during MAE's pixel inference, VitionTS exhibits a strong bias toward *inter-period consistency*, often overshadowing the global trend. Fig. 3 illustrates VisionTS's forecasts on a synthetic sinusoidal time series with a period of 24. As shown in Fig. 3(a)-(d), where the segment length $P$ varies from 24 to 48, forecasts alternate between accurate and inaccurate as $P$ shifts from $1\times$period to $2\times$period, highlighting a strong inductive bias toward periodicity. Notably, the forecasts aren't mere repetitions – the decreasing intra-period amplitude indicates that LVMs can still capture local trends within each period. More quantitative results are deferred to Appendix D.2.

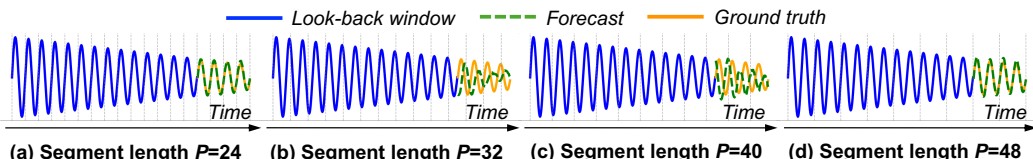

**(a) Segment length *P*=24    (b) Segment length *P*=32    (c) Segment length *P*=40    (d) Segment length *P*=48**

Figure 3: An illustration of LVM forecaster's inductive bias. The time series has a period of 24. The vertical dashed lines mark the segment points. The example indicates a bias towards segment lengths that are multiples of the period in (a)(d) over other segment lengths in (b)(c).

Motivated by this observation, we design the DMMV framework to leverage the inductive bias of LVMs while addressing their limitations. Specifically, the *visual forecaster* $f_{\text{vis}}(\cdot)$ (*i.e.*, LVM) focuses on capturing periodic patterns from the *visual view*, while the *numerical forecaster* $f_{\text{num}}(\cdot)$ models global trends from the *numerical view*, resulting in more balanced forecasting. Fig. 1 presents the two DMMV variants – DMMV-S and DMMV-A – within a decomposition-based architecture. Unlike prior approaches [44, 22, 6, 21], DMMV is explicitly designed to align with the inductive bias of LVMs.

## 3.1   DMMV with Simple Decomposition (DMMV-S)

DMMV-S adopts a simple moving-average (MOV) decomposition [44], which explicitly decomposes an input time series $\mathbf{x}^i$ into a trend part and a seasonal (or periodic) part, reflecting the long-term progression and the seasonality of $\mathbf{x}^i$, respectively. Basically, MOV uses a kernel (*i.e.*, a sliding window) of length $2\lfloor P/2 \rfloor + 1$ to extract the component with frequency lower than $\mathbf{x}^i$'s sampling frequency (*i.e.*, $1/P$), highlighting the global trend. The residual component is the seasonal part. This operation constitutes the decomposition block in Fig. 1(a).

$$\mathbf{x}^i_{\text{trend}} = \texttt{Moving-Average}(\texttt{Padding}(\mathbf{x}^i)), \quad \mathbf{x}^i_{\text{season}} = \mathbf{x}^i - \mathbf{x}^i_{\text{trend}}, \quad 1 \le i \le D \qquad (1)$$

where $\texttt{Padding}(\cdot)$ keeps the length of $\mathbf{x}^i$ fixed.

The visual forecaster $f_{\text{vis}}(\cdot)$ transforms the input $\mathbf{x}^i_{\text{season}}$ into a $224 \times 224 \times 3$ image $\tilde{\mathbf{I}}^i_{\text{season}}$, and outputs the forecast $\hat{\mathbf{y}}^i_{\text{season}} \in \mathbb{R}^H$ for the seasonal component. For the numerical forecaster $f_{\text{num}}(\cdot)$, rather than imposing a specialized inductive bias, we adopt a general-purpose architecture capable of capturing long-term dependencies. We investigate the feasibility of two options and leave other explorations as a future work: (1) A simple linear model motivated by the proven effectiveness of linear methods in LTSF [49, 22, 21], *i.e.*, $\hat{\mathbf{y}}^i_{\text{trend}} = f_{\text{num}}(\mathbf{x}^i_{\text{trend}}) = \mathbf{W}\mathbf{x}^i_{\text{trend}} + \mathbf{b}$, where $\mathbf{W} \in \mathbb{R}^{H \times T}$ and $\mathbf{b} \in \mathbb{R}^H$ are weight and bias, respectively; and (2) A Transformer-based model inspired by PatchTST [30], which segments $\mathbf{x}^i_{\text{trend}}$ into $N$ length-$L$ patches, where $N = \lfloor T/L \rfloor + 1$, to form the input $\mathbf{X}^i_{\text{trend}} \in \mathbb{R}^{L \times N}$, and performs

$$\tilde{\mathbf{X}}^i_{\text{trend}} = \mathbf{W}_{\text{pro}}\mathbf{X}^i_{\text{trend}} + \mathbf{W}_{\text{pos}} \rightarrow \hat{\mathbf{X}}^i_{\text{trend}} = \texttt{Transformer}(\tilde{\mathbf{X}}^i_{\text{trend}}) \rightarrow \hat{\mathbf{y}}^i_{\text{trend}} = \texttt{Linear}(\texttt{Flatten}(\hat{\mathbf{X}}^i_{\text{trend}})) \quad (2)$$

to achieve the forecast $\hat{\mathbf{y}}^i_{\text{trend}} \in \mathbb{R}^H$ for the trend part, where $\mathbf{W}_{\text{pro}} \in \mathbb{R}^{D' \times L}$ is the weight to project the patches to $D'$-dimensional embeddings, $\mathbf{W}_{\text{pos}} \in \mathbb{R}^{D' \times N}$ is a learnable positional encoding, $\texttt{Flatten}(\cdot)$ and $\texttt{Linear}(\cdot)$ are flatten and linear operators.

Finally, $\hat{\mathbf{y}}^i_{\text{season}}$ and $\hat{\mathbf{y}}^i_{\text{trend}}$ are merged to produce the overall forecast $\hat{\mathbf{y}}^i$ for the $i$-th variate. In particular, instead of using the regular summation-based merge, we design an adaptive merge function

with a *light-weight gate* $g = \texttt{sigmoid}(w_g) \in [0,1]$, where $w_g$ is a learnable scalar parameter. To sum up, the overall process of DMMV-S is as follows.

$$\hat{\mathbf{y}}^i = g \circ \hat{\mathbf{y}}^i_{\text{season}} + (1-g) \circ \hat{\mathbf{y}}^i_{\text{trend}}, \quad \text{where} \ \ \hat{\mathbf{y}}^i_{\text{season}} = f_{\text{vis}}(\tilde{\mathbf{I}}^i_{\text{season}}), \ \ \hat{\mathbf{y}}^i_{\text{trend}} = f_{\text{num}}(\mathbf{x}^i_{\text{trend}}) \quad (3)$$

**Remark**. One limitation of DMMV-S is the explicit trend-seasonal decomposition placed on the input $\mathbf{x}^i$, which will enforce $f_{\text{num}}(\cdot)$ and $f_{\text{vis}}(\cdot)$ to fit **pre-defined components** extracted by a certain kernel size. This is not flexible and may not fully leverage LVMs' potential. To address it, we develop DMMV-A to have an adaptive decomposition in the next.

## 3.2 DMMV with Adaptive Decomposition (DMMV-A)

Unlike DMMV-S, DMMV-A *implicitly* decomposes the input $\mathbf{x}^i$ into trend and seasonal components tailored to the strengths of the numerical and visual forecasters, respectively. This is achieved via a *backcast-residual* mechanism (Fig. 1(b)) that leverages LVMs' bias toward periodic patterns. The input $\mathbf{x}^i$ is first transformed into an image $\tilde{\mathbf{I}}^i$ using period-based imaging. Before forecasting, $f_{\text{vis}}(\cdot)$ is used to *backcast* the look-back window by reconstructing masked segments of $\tilde{\mathbf{I}}^i$. An effective masking strategy must: (1) enable full-window reconstruction; (2) align with the forecasting setup (Fig. 2); and (3) minimize the usage of $f_{\text{vis}}(\cdot)$ to avoid computational overhead. To meet these criteria, we propose an efficient *BackCast-Masking* (BCMASK) strategy (Fig. 4), which applies two passes: masking and reconstructing the left and right halves of $\tilde{\mathbf{I}}^i$, respectively.

$$\hat{\mathbf{I}}^i = [\hat{\mathbf{I}}^i_{\text{left}}; \hat{\mathbf{I}}^i_{\text{right}}], \quad \text{with} \ \ \hat{\mathbf{I}}^i_{\text{left}} = f_{\text{vis}}(\tilde{\mathbf{I}}^i_{\text{right}}), \ \ \hat{\mathbf{I}}^i_{\text{right}} = f_{\text{vis}}(\tilde{\mathbf{I}}^i_{\text{left}}) \quad (4)$$

where $\tilde{\mathbf{I}}^i_{\text{right}}$ ($\tilde{\mathbf{I}}^i_{\text{left}}$) is the masked image with right (left) area unmasked, $\hat{\mathbf{I}}^i_{\text{left}}$ ($\hat{\mathbf{I}}^i_{\text{right}}$) is the reconstructed left (right) area, $\hat{\mathbf{I}}^i$ is the reconstruction, or backcast, of $\tilde{\mathbf{I}}^i$ by concatenating $\hat{\mathbf{I}}^i_{\text{left}}$ and $\hat{\mathbf{I}}^i_{\text{right}}$.

BCMASK satisfies all three criteria: (1) it enables full reconstruction of $\tilde{\mathbf{I}}^i$; (2) it uses contiguous segments in $\tilde{\mathbf{I}}^i_{\text{right}}$ ($\tilde{\mathbf{I}}^i_{\text{left}}$) to predict adjacent segments $\hat{\mathbf{I}}^i_{\text{left}}$ ($\hat{\mathbf{I}}^i_{\text{right}}$), mirroring the forecasting process in Fig. 2; and (3) it minimizes the use of $f_{\text{vis}}(\cdot)$ – only two passes are needed, as some unmasked regions of $\tilde{\mathbf{I}}^i$ are required for prediction and must later be masked to complete the full reconstruction.

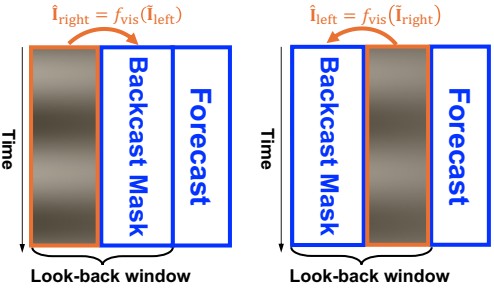

Notably, the backcast in image $\hat{\mathbf{I}}^i$ is biased toward the periodic patterns in $\tilde{\mathbf{I}}^i$. After de-normalization and reverse transformation, a backcast time series $\hat{\mathbf{x}}^i \in \mathbb{R}^T$ is recovered, reflecting periodic compo-

Figure 4: An illustration of BCMASK.

nent in $\mathbf{x}^i$. The residual $\Delta\mathbf{x}^i = \mathbf{x}^i - \hat{\mathbf{x}}^i$ therefore emphasizes the trend. As shown in Fig. 1(b), we feed $\Delta\mathbf{x}^i$ into $f_{\text{num}}(\cdot)$ to produce $\hat{\mathbf{y}}^i_{\text{trend}} \in \mathbb{R}^H$, analogous to its role in DMMV-S. Meanwhile, $f_{\text{vis}}(\cdot)$ predicts from $\tilde{\mathbf{I}}^i$, likely yielding the forecast of seasonal component $\hat{\mathbf{y}}^i_{\text{season}} \in \mathbb{R}^H$. Finally, $\hat{\mathbf{y}}^i_{\text{trend}}$ and $\hat{\mathbf{y}}^i_{\text{season}}$ are fused via the same gating mechanism as Eq. (3). In summary, this defines the overall process of DMMV-A as follows.

$$\hat{\mathbf{y}}^i = g \circ \hat{\mathbf{y}}^i_{\text{season}} + (1-g) \circ \hat{\mathbf{y}}^i_{\text{trend}}, \quad \text{where} \ \ \hat{\mathbf{y}}^i_{\text{season}} = f_{\text{vis}}(\tilde{\mathbf{I}}^i), \ \ \hat{\mathbf{y}}^i_{\text{trend}} = f_{\text{num}}(\Delta\mathbf{x}^i) \quad (5)$$

**Remark**. Unlike DMMV-S, DMMV-A automatically learns a decomposition of $\mathbf{x}^i$ that optimally aligns $f_{\text{num}}(\cdot)$ and $f_{\text{vis}}(\cdot)$ with the forecasting task. As shown in §4.3, this adaptive decomposition effectively separates seasonal and trend components, leveraging the inductive bias of $f_{\text{vis}}(\cdot)$. Unlike the backcast in N-BEATS [31] – designed merely to extract predictive errors – our approach is specifically tailored to exploit LVMs' bias toward periodic patterns, making it fundamentally different.

## 3.3 Model Optimization

After obtaining $\hat{\mathbf{Y}} = [\hat{\mathbf{y}}^1, ..., \hat{\mathbf{y}}^D]^\top \in \mathbb{R}^{D \times H}$, DMMV is trained by minimizing the MSE between $\hat{\mathbf{Y}}$ and $\mathbf{Y}$, *i.e.*, $\frac{1}{D \cdot H} \sum_{i=1}^{D} \sum_{t=1}^{H} \|\hat{\mathbf{Y}}_{it} - \mathbf{Y}_{it}\|_2^2$. As shown in Fig. 1, $f_{\text{num}}(\cdot)$ is trained from scratch, while $f_{\text{vis}}(\cdot)$ uses pre-trained LVM weights with partial fine-tuning. We find that fine-tuning only

the normalization layers yields the best performance, consistent with the findings in [55]. For the choice of LVM, we tested MAE [12] and SimMIM [45], both are self-supervisedly pre-trained LVMs, in §4.2. MAE performs better and is set as the default. Training begins with $f_{\text{vis}}(\cdot)$ frozen while $f_{\text{num}}(\cdot)$ is trained for a number of epochs (*e.g.*, $\sim$30). Then, the norm layers of $f_{\text{vis}}(\cdot)$ are unfrozen and fine-tuned jointly with $f_{\text{num}}(\cdot)$ until convergence or early stopping.

## 4 Experiments

In this section, we compare DMMV with the SOTA methods on LTSF benchmark datasets, and analyze its effectiveness with both quantitative and qualitative studies.

**Datasets**. We adopt 8 widely used MTS benchmarks: ETT (Electricity Transformer Temperature) [53], including ETTh1, ETTh2, ETTm1, ETTm2; Weather [44], Illness [44], Traffic [44], and Electricity [38]. Following standard protocols [44], we split the datasets chronologically into training/validation/test sets using a 60%/20%/20% ratio for ETT and 70%/10%/20% for the others. The prediction horizon $H$ is set to {24, 36, 48, 60} for Illness, and {96, 192, 336, 720} for the remaining datasets. By default, look-back window $T$ is 336. Full dataset details are provided in Appendix B.1.

**The Compared Methods**. We compare DMMV with the SOTA methods, including VLM-based multimodal model: (1) Time-VLM [52]; LVM-based model: (2) VisionTS [4]; LLM-based models: (3) Time-LLM [15], (4) GPT4TS [55], (5) CALF [24]; Transformer-based models: (6) PatchTST [30], (7) FEDformer [54], (8) Autoformer [44], (9) Stationary [25], (10) ETSformer [42], (11) Informer [53]; and non-Transformer models: (12) DLinear [49], (13) TimesNet [43], (14) CycleNet [21]. As we use the standard evaluation protocol, we collect results from prior works: Time-VLM ([52]), LLM-based models (reproduced by [36]), Transformer-based models, PatchTST and TimesNet ([4]), and CycleNet ([21]). Since VisionTS in [4] originally uses dynamic look-back windows such as $T = 1152$ and $T = 2304$ for different datasets, we re-run it with $T = 336$. CycleNet's results on the Illness dataset is unavailable in [21]. Thus we run its official code on the Illness dataset.

We evaluate both DMMV variants – DMMV-S and DMMV-A. A linear forecaster (§3.1) and MAE are set as the default in $f_{\text{num}}(\cdot)$ and $f_{\text{vis}}(\cdot)$, respectively. Ablation studies (§4.2) include variants with a Transformer-based $f_{\text{num}}(\cdot)$ and SimMIM as $f_{\text{vis}}(\cdot)$. Following [4], the imaging period $P$ (§3) for both VisionTS and DMMV is set based on each dataset's sampling frequency (see Appendix B.1). Additional details on all compared methods are in Appendix B.2.

**Evaluation**. Following [30, 49, 36], we use Mean Squared Error (MSE) and Mean Absolute Error (MAE) to evaluate the LTSF performance of the compared methods.

### 4.1 Experimental Results

Table 1 summarizes the LTSF performance of 10 representative methods across four categories: MMV-based, visual-view-based, language-view-based, and numerical-view-based approaches, with full results for all 16 methods provided in Appendix D.1. Time-VLM's results on the Illness dataset are not reported in [52] and its code is unavailable at the time of this experiment, thus are marked by "–". For DMMV, the stronger variant, DMMV-A, is reported. In Table 1, several key insights emerge: (1) MMV and visual-view methods generally outperform language-view methods, underscoring the effectiveness of LVMs, particularly when integrated within MMV frameworks; (2) Numerical-view models such as PatchTST and CycleNet remain competitive, especially on datasets where VisionTS underperforms (*e.g.*, ETTm2 and Electricity), highlighting their potential to complement visual models; (3) The strong results of CycleNet, a lightweight model with learnable decomposition, demonstrate the value of combining simplicity with structure in LTSF; (4) Notably, DMMV-A, which unifies visual and numerical views through a novel adaptive decomposition, outperforms the baselines in most cases, achieving 43 first-places and confirming its effectiveness; (5) Lastly, while VisionTS performs well on highly periodic datasets (*e.g.*, ETTh1, ETTm1, Traffic) due to MAE's inductive bias toward periodicity, DMMV-A alleviates this bias, resulting in more generalizable forecasts.

Fig. 5 presents critical difference (CD) diagrams [11] showing the average rank of all 16 methods based on MSE and MAE across prediction lengths and all datasets. DMMV-S ranks 4.5/16 in MSE and 7.1/16 in MAE, underscoring the benefit of the adaptive decomposition used in DMMV-A (Fig. 1(b)). From Fig. 5, DMMV-S's comparable ranks to CycleNet indicate even with a simpler, fixed decomposition, DMMV-S exhibits an ability that a strong SOTA model with learnable decomposition has. In §4.3, a detailed comparison between DMMV-S and DMMV-A is provided.

Table 1: LTSF performance comparison on the benchmark datasets. Lower MSE and MAE indicate better performance. **Red** values indicate the best MSE and MAE per row. `Time-VLM`'s results on the Illness dataset are unavailable in [52] and its code was unavailable at the time of this experiment.

| View | Multi-Modal | | | | Visual | | Language | | | | Numerical | | | | | | | | | |
|---|---|---|---|---|---|---|---|---|---|---|---|---|---|---|---|---|---|---|---|---|
| **Model** | DMMV-A | | Time-VLM | | VisionTS | | GPT4TS | | Time-LLM | | PatchTST | | CycleNet | | TimesNet | | DLinear | | FEDformer | |
| **Metric** | MSE | MAE | MSE | MAE | MSE | MAE | MSE | MAE | MSE | MAE | MSE | MAE | MSE | MAE | MSE | MAE | MSE | MAE | MSE | MAE |
| ETTh1 96 | **0.354** | 0.389 | 0.361 | **0.386** | 0.355 | **0.386** | 0.370 | 0.389 | 0.376 | 0.402 | 0.370 | 0.399 | 0.374 | 0.396 | 0.384 | 0.402 | 0.375 | 0.399 | 0.376 | 0.419 |
| ETTh1 192 | **0.393** | **0.405** | 0.397 | 0.415 | 0.395 | 0.407 | 0.412 | 0.413 | 0.407 | 0.421 | 0.413 | 0.421 | 0.406 | 0.415 | 0.436 | 0.429 | 0.405 | 0.416 | 0.420 | 0.448 |
| ETTh1 336 | **0.387** | **0.413** | 0.420 | 0.421 | 0.419 | 0.421 | 0.448 | 0.431 | 0.430 | 0.438 | 0.422 | 0.436 | 0.431 | 0.430 | 0.491 | 0.469 | 0.439 | 0.416 | 0.459 | 0.465 |
| ETTh1 720 | 0.445 | 0.450 | **0.441** | 0.458 | 0.458 | 0.460 | **0.441** | **0.449** | 0.457 | 0.468 | 0.447 | 0.466 | 0.450 | 0.464 | 0.521 | 0.500 | 0.472 | 0.490 | 0.506 | 0.507 |
| ETTh1 Avg. | **0.395** | **0.414** | 0.405 | 0.420 | 0.407 | 0.419 | 0.418 | 0.421 | 0.418 | 0.432 | 0.413 | 0.431 | 0.415 | 0.426 | 0.458 | 0.450 | 0.423 | 0.430 | 0.440 | 0.460 |
| ETTh2 96 | 0.294 | 0.349 | **0.267** | 0.335 | 0.288 | **0.334** | 0.280 | 0.335 | 0.286 | 0.346 | 0.274 | 0.336 | 0.279 | 0.341 | 0.340 | 0.374 | 0.289 | 0.353 | 0.358 | 0.397 |
| ETTh2 192 | 0.339 | 0.395 | **0.326** | **0.373** | 0.349 | 0.380 | 0.348 | 0.380 | 0.361 | 0.391 | 0.339 | 0.379 | 0.342 | 0.385 | 0.402 | 0.414 | 0.383 | 0.418 | 0.429 | 0.439 |
| ETTh2 336 | **0.322** | 0.384 | 0.357 | 0.406 | 0.364 | 0.398 | 0.380 | 0.405 | 0.390 | 0.414 | 0.329 | **0.380** | 0.371 | 0.413 | 0.452 | 0.452 | 0.448 | 0.465 | 0.496 | 0.487 |
| ETTh2 720 | 0.392 | 0.425 | 0.412 | 0.449 | 0.403 | 0.431 | 0.406 | 0.436 | 0.405 | 0.434 | **0.379** | **0.422** | 0.426 | 0.451 | 0.462 | 0.468 | 0.605 | 0.551 | 0.463 | 0.474 |
| ETTh2 Avg. | 0.337 | 0.388 | 0.341 | 0.391 | 0.351 | 0.386 | 0.354 | 0.389 | 0.361 | 0.396 | **0.330** | **0.379** | 0.355 | 0.398 | 0.414 | 0.427 | 0.431 | 0.447 | 0.437 | 0.449 |
| ETTm1 96 | **0.279** | **0.329** | 0.304 | 0.346 | 0.284 | 0.332 | 0.300 | 0.340 | 0.291 | 0.341 | 0.290 | 0.342 | 0.299 | 0.348 | 0.338 | 0.375 | 0.299 | 0.343 | 0.379 | 0.419 |
| ETTm1 192 | **0.317** | **0.357** | 0.332 | 0.366 | 0.327 | 0.362 | 0.343 | 0.368 | 0.341 | 0.369 | 0.332 | 0.369 | 0.334 | 0.367 | 0.374 | 0.387 | 0.335 | 0.365 | 0.426 | 0.441 |
| ETTm1 336 | **0.351** | 0.381 | 0.364 | 0.383 | 0.354 | 0.382 | 0.376 | 0.382 | 0.359 | **0.379** | 0.366 | 0.392 | 0.368 | 0.386 | 0.410 | 0.411 | 0.369 | 0.386 | 0.445 | 0.459 |
| ETTm1 720 | 0.411 | 0.415 | **0.402** | **0.410** | 0.411 | 0.415 | 0.431 | 0.416 | 0.433 | 0.419 | 0.416 | 0.420 | 0.417 | 0.414 | 0.478 | 0.450 | 0.425 | 0.421 | 0.543 | 0.490 |
| ETTm1 Avg. | **0.340** | **0.371** | 0.351 | 0.376 | 0.344 | 0.373 | 0.363 | 0.378 | 0.356 | 0.377 | 0.351 | 0.381 | 0.355 | 0.379 | 0.400 | 0.406 | 0.357 | 0.379 | 0.448 | 0.452 |
| ETTm2 96 | 0.172 | 0.260 | 0.160 | 0.250 | 0.174 | 0.262 | 0.163 | 0.249 | 0.162 | 0.248 | 0.165 | 0.255 | **0.159** | **0.247** | 0.187 | 0.267 | 0.167 | 0.260 | 0.203 | 0.287 |
| ETTm2 192 | 0.227 | 0.298 | 0.215 | 0.291 | 0.228 | 0.297 | 0.222 | 0.291 | 0.235 | 0.304 | 0.220 | 0.292 | **0.214** | **0.286** | 0.249 | 0.309 | 0.224 | 0.303 | 0.269 | 0.328 |
| ETTm2 336 | 0.272 | 0.327 | 0.270 | 0.325 | 0.281 | 0.337 | 0.273 | 0.327 | 0.280 | 0.329 | 0.274 | 0.329 | **0.269** | **0.322** | 0.321 | 0.351 | 0.281 | 0.342 | 0.325 | 0.366 |
| ETTm2 720 | 0.351 | 0.381 | **0.348** | 0.378 | 0.384 | 0.410 | 0.357 | **0.376** | 0.366 | 0.382 | 0.362 | 0.385 | 0.363 | 0.382 | 0.408 | 0.403 | 0.397 | 0.421 | 0.421 | 0.415 |
| ETTm2 Avg. | 0.256 | 0.317 | **0.248** | 0.311 | 0.267 | 0.327 | 0.254 | 0.311 | 0.261 | 0.316 | 0.255 | 0.315 | 0.251 | **0.309** | 0.291 | 0.333 | 0.267 | 0.332 | 0.305 | 0.349 |
| Illness 24 | 1.409 | **0.754** | – | – | 1.613 | 0.834 | 1.869 | 0.823 | 1.792 | 0.807 | **1.319** | **0.754** | 2.255 | 1.017 | 2.317 | 0.934 | 2.215 | 1.081 | 3.228 | 1.260 |
| Illness 36 | **1.290** | **0.745** | – | – | 1.316 | 0.750 | 1.853 | 0.854 | 1.833 | 0.833 | 1.430 | 0.834 | 2.121 | 0.950 | 1.972 | 0.920 | 1.963 | 0.963 | 2.679 | 1.080 |
| Illness 48 | **1.499** | **0.810** | – | – | 1.548 | 0.818 | 1.886 | 0.855 | 2.269 | 1.012 | 1.553 | 0.815 | 2.187 | 1.007 | 2.238 | 0.940 | 2.130 | 1.024 | 2.622 | 1.078 |
| Illness 60 | **1.428** | **0.773** | – | – | 1.450 | 0.783 | 1.877 | 0.877 | 2.177 | 0.925 | 1.470 | 0.788 | 2.185 | 0.997 | 2.027 | 0.928 | 2.368 | 1.096 | 2.857 | 1.157 |
| Illness Avg. | **1.407** | **0.771** | – | – | 1.482 | 0.796 | 1.871 | 0.852 | 2.018 | 0.894 | 1.443 | 0.798 | 2.187 | 0.992 | 2.139 | 0.931 | 2.169 | 1.041 | 2.847 | 1.144 |
| Electricity 96 | **0.126** | **0.213** | 0.142 | 0.245 | 0.127 | 0.217 | 0.141 | 0.239 | 0.137 | 0.233 | 0.129 | 0.222 | 0.128 | 0.223 | 0.168 | 0.272 | 0.140 | 0.237 | 0.193 | 0.308 |
| Electricity 192 | 0.145 | **0.237** | 0.157 | 0.260 | 0.148 | **0.237** | 0.158 | 0.253 | 0.152 | 0.247 | 0.157 | 0.240 | **0.144** | **0.237** | 0.184 | 0.289 | 0.153 | 0.249 | 0.201 | 0.315 |
| Electricity 336 | 0.162 | 0.254 | 0.174 | 0.276 | 0.163 | **0.253** | 0.172 | 0.266 | 0.169 | 0.267 | 0.163 | 0.259 | **0.160** | 0.254 | 0.220 | 0.300 | 0.169 | 0.267 | 0.214 | 0.329 |
| Electricity 720 | **0.197** | **0.286** | 0.214 | 0.308 | 0.199 | 0.293 | 0.207 | 0.293 | 0.200 | 0.290 | **0.197** | 0.290 | 0.198 | 0.287 | 0.220 | 0.320 | 0.203 | 0.301 | 0.246 | 0.355 |
| Electricity Avg. | **0.158** | **0.248** | 0.172 | 0.272 | 0.159 | 0.250 | 0.170 | 0.263 | 0.165 | 0.259 | 0.162 | 0.253 | **0.158** | 0.250 | 0.193 | 0.295 | 0.166 | 0.264 | 0.214 | 0.327 |
| Weather 96 | **0.143** | 0.195 | 0.148 | 0.200 | 0.146 | 0.191 | 0.148 | **0.188** | 0.155 | 0.199 | 0.149 | 0.198 | 0.167 | 0.221 | 0.172 | 0.220 | 0.176 | 0.237 | 0.217 | 0.296 |
| Weather 192 | **0.187** | 0.242 | 0.193 | 0.240 | 0.194 | 0.238 | 0.192 | **0.230** | 0.223 | 0.261 | 0.194 | 0.241 | 0.212 | 0.258 | 0.219 | 0.261 | 0.220 | 0.282 | 0.276 | 0.336 |
| Weather 336 | **0.237** | **0.273** | 0.243 | 0.281 | 0.243 | 0.275 | 0.246 | **0.273** | 0.251 | 0.279 | 0.245 | 0.282 | 0.260 | 0.293 | 0.280 | 0.306 | 0.265 | 0.319 | 0.339 | 0.380 |
| Weather 720 | **0.302** | **0.315** | 0.312 | 0.332 | 0.318 | 0.328 | 0.320 | 0.328 | 0.345 | 0.342 | 0.328 | 0.334 | 0.328 | 0.339 | 0.365 | 0.359 | 0.333 | 0.362 | 0.403 | 0.428 |
| Weather Avg. | **0.217** | 0.256 | 0.224 | 0.263 | 0.225 | 0.258 | 0.227 | **0.255** | 0.244 | 0.270 | 0.226 | 0.264 | 0.242 | 0.278 | 0.259 | 0.287 | 0.249 | 0.300 | 0.309 | 0.360 |
| Traffic 96 | **0.344** | 0.237 | 0.393 | 0.290 | 0.346 | **0.232** | 0.396 | 0.264 | 0.392 | 0.267 | 0.360 | 0.249 | 0.397 | 0.278 | 0.593 | 0.321 | 0.410 | 0.282 | 0.587 | 0.366 |
| Traffic 192 | **0.363** | 0.249 | 0.405 | 0.296 | 0.376 | **0.245** | 0.412 | 0.268 | 0.409 | 0.271 | 0.379 | 0.256 | 0.411 | 0.283 | 0.617 | 0.336 | 0.423 | 0.287 | 0.604 | 0.373 |
| Traffic 336 | **0.387** | 0.256 | 0.420 | 0.305 | 0.389 | **0.252** | 0.421 | 0.273 | 0.434 | 0.296 | 0.392 | 0.264 | 0.424 | 0.289 | 0.629 | 0.336 | 0.436 | 0.296 | 0.621 | 0.383 |
| Traffic 720 | 0.433 | **0.284** | 0.459 | 0.323 | **0.432** | 0.293 | 0.455 | 0.291 | 0.451 | 0.291 | **0.432** | 0.286 | 0.450 | 0.305 | 0.640 | 0.350 | 0.466 | 0.315 | 0.626 | 0.382 |
| Traffic Avg. | **0.382** | 0.257 | 0.419 | 0.304 | 0.386 | **0.256** | 0.421 | 0.274 | 0.422 | 0.281 | 0.391 | 0.264 | 0.421 | 0.289 | 0.620 | 0.336 | 0.434 | 0.295 | 0.610 | 0.376 |
| **# Wins** | **43** | | 9 | | 9 | | 7 | | 1 | | 9 | | 11 | | 0 | | 0 | | 0 | |

**(a) MSE Ranking**                    **(b) MAE Ranking**

Figure 5: Critical difference (CD) diagram on the average rank of all 16 compared methods in terms of (a) MSE and (b) MAE over all benchmark datasets. The lower rank (left of the scale) is better.

## 4.2 Ablation Analysis

We validate the design of DMMV-A through ablation studies on four datasets; DMMV-S results are deferred to Appendix D.3 for brevity. Table 2 summarizes the analysis: (a) replaces the linear model in $f_{num}(\cdot)$ with a `PatchTST`-style `Transformer` (see §3.1); (b) swaps `MAE` with `SimMIM` [45] as $f_{vis}(\cdot)$; (c) replaces the gating fusion with a simple sum; (d) removes BCMASK, performing backcasting and forecasting on the full, unmasked look-back window; (e) substitutes BCMASK with random masking; (f) freezes the entire $f_{vis}(\cdot)$ instead of fine-tuning norm layers; and (g) removes the backcast-residual mechanism, feeding both $f_{num}(\cdot)$ and $f_{vis}(\cdot)$ the same input $\mathbf{x}^i$ and merging their outputs via gating.

Table 2 reveals key insights into DMMV-A's design. In (a), replacing the linear numerical forecaster with a `Transformer` slightly degrades performance, likely due to the increased difficulty of jointly training `Transformer` with LVMs. In (b), `MAE` outperforms `SimMIM` as $f_{vis}(\cdot)$, likely due to its `ViT`-based reconstruction decoder being better suited for pixel-level tasks like LTSF than `SimMIM`'s linear decoder, while both models share similar encoder architectures. In (c), gate-based fusion

Table 2: Ablation analysis of DMMV-A. MSE and MAE are averaged over different prediction lengths. Lower MSE and MAE are better. "Improvement" of each ablation is relative to DMMV-A.

| Dataset (→) | ETTh1 | | ETTm1 | | Illness | | Weather | |
|---|---|---|---|---|---|---|---|---|
| Method (↓), Metric (→) | MSE | MAE | MSE | MAE | MSE | MAE | MSE | MAE |
| DMMV-A | 0.395 | **0.414** | 0.340 | **0.371** | **1.407** | **0.771** | **0.217** | **0.256** |
| (a) $f_{num}(\cdot) \rightarrow$ Transformer | 0.407 | 0.421 | 0.339 | 0.372 | 1.442 | 0.786 | 0.219 | 0.260 |
| Improvement | -3.04% | -1.69% | +0.29% | -0.27% | -2.49% | -1.95% | -0.92% | -1.56% |
| (b) $f_{vis}(\cdot) \rightarrow$ SimMIM | 0.407 | 0.415 | 0.345 | 0.377 | 1.649 | 0.814 | 0.227 | 0.261 |
| Improvement | -3.04% | -0.24% | -1.47% | -1.62% | -17.20% | -5.58% | -4.61% | -1.95% |
| (c) Gate $\rightarrow$ Sum | 0.414 | 0.427 | 0.352 | 0.383 | 1.606 | 0.863 | 0.233 | 0.278 |
| Improvement | -4.81% | -3.14% | -3.53% | -3.23% | -14.14% | -11.93% | -7.37% | -8.59% |
| (d) BCMASK $\rightarrow$ No mask | 0.426 | 0.441 | 0.349 | 0.377 | 1.493 | 0.828 | 0.221 | 0.267 |
| Improvement | -7.85% | -6.52% | -2.65% | -1.62% | -6.11% | -7.39% | -1.84% | -4.30% |
| (e) BCMASK $\rightarrow$ Random mask | **0.394** | **0.414** | 0.340 | 0.372 | 1.472 | 0.829 | 0.223 | 0.262 |
| Improvement | 0.25% | 0.00% | 0.00% | -0.27% | -4.62% | -7.52% | -2.76% | -2.34% |
| (f) Freeze $f_{vis}(\cdot)$ | 0.431 | 0.428 | 0.358 | 0.380 | 1.442 | 0.773 | 0.246 | 0.288 |
| Improvement | -9.11% | -3.38% | -5.29% | -2.43% | -2.49% | -0.26% | -13.36% | -12.50% |
| (g) W/o decomposition | 0.408 | 0.424 | **0.338** | 0.373 | 1.712 | 0.903 | 0.219 | 0.268 |
| Improvement | -3.29% | -2.42% | 0.59% | -0.54% | -21.68% | -17.12% | -0.92% | -4.69% |

Figure 6: Comparing DMMV-S and DMMV-A *w.r.t.* gate weights on visual and numerical forecasters.

outperforms simple summation, highlighting its adaptability to the distinct outputs of $f_{num}(\cdot)$ and $f_{vis}(\cdot)$. (d) and (e) underscore the importance of BCMASK: removing it (*i.e.*, (d) "No mask") recovers the full look-back window as the backcasted seasonal component, diminishing the trend signal and weakening $f_{num}(\cdot)$, while "Random mask" (*i.e.*, (e)) performs slightly worse due to poorer periodic pattern extraction, which leads to many fluctuations. In §4.3, we provide visual examples to compare these masking strategies. In (f), fine-tuning only the norm layers significantly improves performance over freezing, confirming the benefit of coordinated learning between forecasters, as described in §3.3. Finally, (g) shows that removing the backcast-residual mechanism causes a major performance drop, affirming its role in effective decomposition. Overall, the LVM decoder, fusion strategy, masking method, training approach, and decomposition mechanism are crucial to DMMV-A's success.

## 4.3 Performance Analysis

In this section, we perform an in-depth analysis of DMMV using the same four datasets as in §4.2.

**The Difference between DMMV-S and DMMV-A.** Fig. 5 highlights DMMV-A's superiority over DMMV-S, largely due to its adaptive decomposition mechanism. A key advantage of the gate-based fusion is its interpretability. As shown in Fig. 6, which presents average gate weights across datasets, DMMV-A consistently places more weight on $f_{vis}(\cdot)$, while DMMV-S tends to balance both $f_{num}(\cdot)$ and $f_{vis}(\cdot)$ but leans toward $f_{num}(\cdot)$. In DMMV-A, these weights are learned based on forecasting performance, emphasizing $f_{vis}(\cdot)$'s importance. Notably, although $f_{num}(\cdot)$ receives less weight, it remains essential – as evidenced by DMMV-A outperforming the visual-only baseline VisionTS in Table 1. In contrast, DMMV-S's weights are limited by its fixed moving-average decomposition, leading to a non-adaptive and suboptimal allocation of forecasting roles.

Fig. 7 provides example decompositions by DMMV-S and DMMV-A (additional cases in Appendix D.4). DMMV-A produces a smooth, clearly periodic component – consistent with expectations, and a trend component with some noises. In contrast, DMMV-S's moving-average yields a smoother trend by absorbing fluctuations, pushing noise into the seasonal component. This makes forecasting harder for $f_{vis}(\cdot)$, which is more sensitive to fluctuations than $f_{num}(\cdot)$, resulting in lower weights of $f_{vis}(\cdot)$ in Fig. 6. Since periodic patterns are crucial for long-term forecasting, as identified by [22, 21], the clearer period separation in DMMV-A leads to forecasts that better match the ground truth.

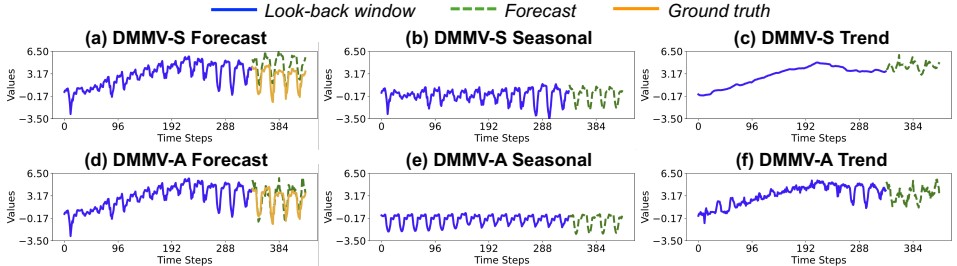

Figure 7: The decompositions of DMMV-S and DMMV-A on the same example in ETTh1: (a)(d) input time series and forecasts, (b)(e) seasonal component, and (c)(f) trend component.

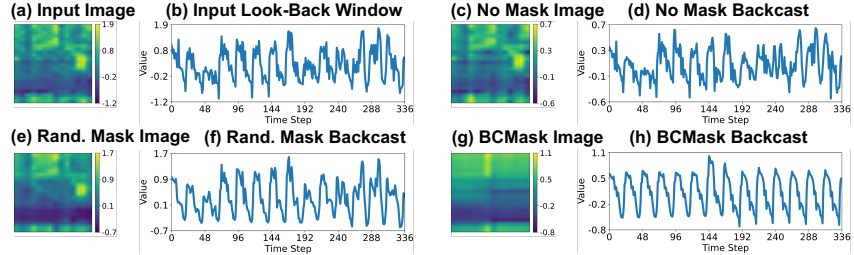

Figure 8: Comparison of different masking methods on the same example in ETTh1. (a) image of input look-back window; (c)(e)(g) are images of backcast output by DMMV-A: (c) uses "No mask"; (e) uses "Random mask"; (g) uses BCMASK. (b)(d)(f)(h) are their recovered time series, respectively.

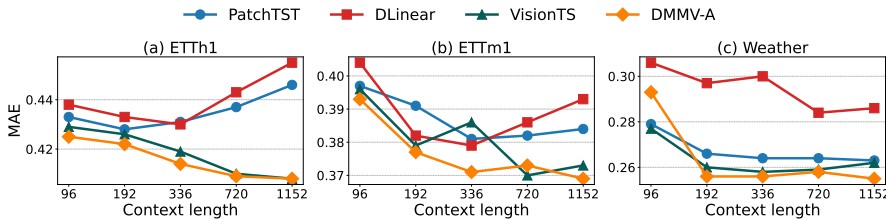

Figure 9: Average MAE comparison with varying look-back window (or context) lengths.

**The Effectiveness of BCMASK.** Fig. 8 compares the backcast results of DMMV-A using BCMASK, "No mask", and "Random mask" (as in Table 2) on a sample case; more examples are in Appendix D.5. BCMASK produces a smooth image along the temporal ($x$-axis) segments, effectively capturing clean periodic patterns. In contrast, "No mask" closely replicates the input, offering no meaningful decomposition. "Random mask" performs moderately well, resembling BCMASK but with less temporal smoothness, indicating a less optimal decomposition.

**Impact of Look-Back Window.** Fig. 9 compares DMMV-A with a visual forecaster (`VisionTS`) and two numerical forecasters (`PatchTST`, `DLinear`), which can serve as its single-view ablations. Illness dataset is excluded due to its short time series (966 time steps). Using MAE metric (MSE results in Appendix D.6), we observe that DMMV-A and `VisionTS` benefit from longer look-back windows, while `PatchTST` and `DLinear` degrade beyond a length of 336. Notably, DMMV-A outperforms `VisionTS` at length 1152, highlighting the advantage of explicitly modeling global trends.

## 5  Conclusion

This paper introduces DMMV, a novel MMV framework that leverages LVMs and adaptive decomposition to enhance LTSF. By addressing the inductive bias of LVMs toward periodicity through a tailored backcast-residual decomposition, DMMV effectively integrates numerical and visual perspectives. Extensive experiments on benchmark datasets demonstrate that DMMV outperforms both single-view and SOTA multi-modal baselines, validating its effectiveness. This work highlights the potential of MMVs and LVMs in advancing LTSF, offering a new direction for future research in this domain.

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

## A    Limitation and Broader Impact

### A.1    Discussion of Limitations

Adapting LVMs to LTSF is an emerging area of active research. This work serves a pioneering effort in investigating LVMs within an MMV framework for LTSF. As an initial exploration, we acknowledge some limitations in this work. First, one limitation of the current best LVM forecaster (*e.g.*, VisionTS) is its sensitivity to segment length used in image construction due to its inductive bias, as discussed in §3 (Fig. 3). By incorporating an additional numerical view for modeling the global trend, the proposed DMMV is expected to alleviate this sensitivity. In our further analysis in Appendix D.2, we observe DMMV-A is less sensitive to the change of segment lengths than VisionTS on some datasets, but cannot consistently enhance the robustness over different datasets, despite the improved overall forecasting performance. This may be caused by the higher weights automatically allocated to $f_{\text{vis}}(\cdot)$ than $f_{\text{num}}(\cdot)$ by the gate fusion mechanism (Fig. 6), which could make the model prone to inherit the behavior of the LVM used in $f_{\text{vis}}(\cdot)$ to some extent, including its sensitivity to segment length, but with a less extent than a sole LVM. As such, a future work to improve DMMV is to reduce such sensitivity to an unnoticeable effect. Second, under our proposed BCMASK strategy, the vision backbone in $f_{\text{vis}}(\cdot)$ is reused three times during training and inference – once for forecasting and twice for reconstructing different masked parts of the look-back window. The triple use of $f_{\text{vis}}(\cdot)$ could lead to a non-trivial computational overhead. In this work, considering the remarkable performance improvement of BCMASK over other masking strategies (Table 2) and its possibly minimum use of $f_{\text{vis}}(\cdot)$, as analyzed in §3.2, we take it as the current solution. However, as a future work, we expect to further reduce the use of $f_{\text{vis}}(\cdot)$ and improve the efficiency, by methods such as joint backcasting and forecasting within a single forward pass or amortizing the use of LVMs across multiple time series samples. Finally, the proposed method shares a similar limitation of the existing LVM forecasters. The imaging process transforms an input time series into a 2D image-like representation to fit pre-trained LVMs, which typically expect a high-resolution input of size such as $224 \times 224 \times 3$. Therefore, upsampling is performed on a smaller image obtained from the patched time series to fit the input requirements of LVMs. While critical to compatibility, resizing may introduce subtle changes that may distort the original temporal structures to some extent. Thus an imaging process that can reflect the temporal patterns more accurately is in demand in future works.

### A.2    Broader Impact

LTSF plays a vital role across various domains, including geoscience [1], neuroscience [3], energy [18], healthcare [28], and smart city [27]. This work proposes a novel MMV framework DMMV that integrates LVMs and a numerical forecaster, which could serve as a groundwork in the emerging area of LVM-based time series analysis and shed some lights on broader areas that integrate LLMs, VLMs, and large multi-modal models (LMMs) for future research on multi-modal and agentic time series analysis. This work does not involve sensitive data, legal risks, or ethical concerns. To the best of our knowledge, it does not adversely affect any specific population. The proposed method could serve as a general-purpose time series forecasting technique with a relatively broad applicability and social acceptability.

## B    Benchmark and Baseline

### B.1    Benchmark Datasets

Following [53, 44, 30, 49, 36, 4], our experiments are conducted on 8 widely used LTSF benchmark datasets that cover a wide range of sampling frequencies, number of variates, levels of periodicity, and real-world domains. The four ETT datasets (ETTh1, ETTh2, ETTm1, ETTm2) record oil temperature from two electric transformers, sampled at 15-minute and hourly intervals. The Weather dataset collects measurements of meteorological indicators in Germany every 10 minutes. The Illness dataset keeps weekly counts of patients and the influenza-like illness ratio from the United States. The Traffic dataset measures hourly road occupancy rates from sensors on San Francisco freeways. The Electricity dataset records hourly electricity consumption of Portuguese clients. Table 3 summarizes the statistics of the datasets.

Table 3: Statistics of the benchmark datasets. "Dataset Size" is organized in (Train, Validation, Test).

| Dataset | # Variates | Series Length | Dataset Size | Frequency |
|---|---|---|---|---|
| ETTh1 | 7 | 17420 | (8545, 2881, 2881) | Hourly |
| ETTh2 | 7 | 17420 | (8545, 2881, 2881) | Hourly |
| ETTm1 | 7 | 69680 | (34465, 11521, 11521) | 15 mins |
| ETTm2 | 7 | 69680 | (34465, 11521, 11521) | 15 mins |
| Weather | 321 | 52696 | (36792, 5271, 10540) | 10 mins |
| Illness | 7 | 966 | (617, 74, 170) | Weekly |
| Traffic | 862 | 17544 | (12185, 1757, 3509) | Hourly |
| Electricity | 21 | 26304 | (18317, 2633, 5261) | Hourly |

## B.2 Baselines

In the following, we provide a brief description for each baseline method involved in our experiments.

- `Time-VLM` [52] integrates time series data with visual views and contextual texts using a pre-trained VLM, `ViLT`, to enhance forecasting performance.
- `VisionTS` [4] reformulates time series forecasting as an image reconstruction problem using an LVM, `MAE`, for zero/few/full-shot forecasting.
- `Time-LLM` [15] reprograms LLMs by aligning time series patches with text tokens, enabling time series forecasting without re-training LLMs.
- `GPT4TS` [55] demonstrates that frozen pretrained LLMs, *e.g.*, `GPT`, can be directly applied to a variety of time series tasks with strong performance.
- `CALF` [24] adapts LLMs to time series forecasting via cross-modal fine-tuning, bridging the distribution gap between textual and temporal data.
- `CycleNet` [21] enhances LTSF by explicitly modeling the periodic patterns in time series through a residual cycle forecasting technique.
- `PatchTST` [30] introduces a patching strategy and a channel-independence strategy for LTSF. It uses patches of time series as the input to a `Transformer` to capture the temporal dependency of semantically meaningful tokens (*i.e.*, patches).
- `TimesNet` [43] transforms an input time series into a 2D image-like representation and models temporal variations in the image using inception-like blocks for time series analysis.
- `DLinear` [49] decomposes an input time series into trend and seasonal components, each of which is modeled by linear layers for time series forecasting.
- `FEDformer` [54] incorporates frequency-enhanced attention mechanisms by combining Fourier transforms with seasonal-trend decomposition in a `Transformer` framework.
- `Autoformer` [44] introduces an auto-correlation mechanism within a `Transformer` architecture to capture long-term dependencies in time series data.
- `Stationary` [25] combines series stationarization and de-stationary attention mechanisms to solve the over-stationarization problem in time series forecasting.
- `ETSformer` [42] decomposes an input time series into interpretable components with exponential smoothing attention and frequency attention for time series forecasting.
- `Informer` [53] proposes a ProbSparse self-attention mechanism to reduce the computational complexity of LTSF with `Transformer` models.

## C  Implementation Details

### C.1  Pre-trained LVM Checkpoints

As described in §3.3, $f_{\text{vis}}(\cdot)$ uses pre-trained LVMs. For `MAE`, we use the checkpoint released by *Meta Research* [1], which was pretrained on $224 \times 224 \times 3$ sized images from *ImageNet-1K* [5] with

---

[1] https://github.com/facebookresearch/mae

**Algorithm 1:** The Training Algorithm of DMMV-A

**Input:** training dataset $\mathcal{D}_{\text{train}} = \{\mathbf{X}_i, \mathbf{Y}_i\}_{i=1}^n$, where $\mathbf{X}_i \in \mathbb{R}^{D \times T}$ is an MTS, $\mathbf{Y}_i \in \mathbb{R}^{D \times H}$ is the ground truth of forecast
**Output:** model parameters of DMMV-A

1   Load pre-trained $f_{\text{vis}}(\cdot)$ and freeze its weights
2   Randomly initialize $f_{\text{num}}(\cdot)$ and the gating parameter $g$

    /* stage 1:   numerical forecaster training */
3   **for** $i \leftarrow 1$ *to MaxEpoch* **do**
4     **for** $(\mathbf{X}, \mathbf{Y})$ *in Dataloader*$(\mathcal{D}_{train})$ **do**
       /* channel-independence strategy is applied in the following */
5        $\hat{\mathbf{X}}_{\text{season}}, \hat{\mathbf{Y}}_{\text{season}} \leftarrow f_{\text{vis}}(\mathbf{X}, \text{BCMASK})$      // backcast/forecast seasonal part
6        $\mathbf{X}_{\text{trend}} \leftarrow \mathbf{X} - \hat{\mathbf{X}}_{\text{season}}$           // extract trend component
7        $\hat{\mathbf{Y}}_{\text{trend}} \leftarrow f_{\text{num}}(\mathbf{X}_{\text{trend}})$          // forecast trend with $f_{\text{num}}(\cdot)$
8        $\hat{\mathbf{Y}} \leftarrow g \circ \hat{\mathbf{Y}}_{\text{season}} + (1-g) \circ \hat{\mathbf{Y}}_{\text{trend}}$          // gate fusion
9        Calculate $\ell_{\text{MSE}}(\hat{\mathbf{Y}}, \mathbf{Y})$      // calculate MSE loss as specified in §3.3
10      Update model parameters of $f_{\text{num}}(\cdot)$ and $g$
11      **if** *Early stopping condition is TRUE* **then**
12        Break
13      **end**
14    **end**
15   **end**

    /* stage 2:   joint training */
16   Unfreeze the norm layers in $f_{\text{vis}}(\cdot)$
17   **for** $i \leftarrow 1$ *to MaxEpoch* **do**
18    **for** $(\mathbf{X}, \mathbf{Y})$ *in Dataloader*$(\mathcal{D}_{train})$ **do**
       /* Repeat lines 5-9 */
19      Update model parameters of $f_{\text{num}}(\cdot)$, norm layers in $f_{\text{vis}}(\cdot)$, and parameter $g$
20      **if** *Early stopping condition is TRUE* **then**
21        Break
22      **end**
23    **end**
24   **end**

`ViT-Base` Backbone. For `SimMIM`, we adopt the checkpoint released by *Microsoft*[2], which has the same pretraining setting as aforementioned for `MAE`. For these two LVM backbones, the base versions are adopted to balance the performance and computational costs.

### C.2   Training Details

For training the proposed DMMV-S and DMMV-A models, we adopt AdamW optimizer throughout the experiments. The batch size is set to 64 for the ETT datasets and Illness dataset, and set to 8 for the other three datasets to balance training stability and memory consumption.

For both DMMV-S and DMMV-A, we propose a two-stage training scheme to facilitate effective integration of numerical and visual features:

- **Stage 1** (Numerical forecaster training). In this stage, we freeze all parameters of $f_{\text{vis}}(\cdot)$ and train $f_{\text{num}}(\cdot)$ only. This warm-up step prevents $f_{\text{vis}}(\cdot)$ from updating with unstable gradients caused by the random representations from the under-trained $f_{\text{num}}(\cdot)$. In this stage, the learning rate is set to 0.01. The training runs up to a maximum of 50 epochs on the training set. Early stopping is applied with a patience of 10 epochs.

---

[2] https://github.com/microsoft/SimMIM

- **Stage 2** (Joint training). In this stage, we unfreeze the layer normalization parameter in $f_{\text{vis}}(\cdot)$ and jointly train them with $f_{\text{num}}(\cdot)$ to enable deep fusion of visual and numerical views. The learning rate is reduced to 0.005 to preserve learned features and stabilize training. The training at this stage runs up to 5 epochs. Early stopping is applied with a patience of 2 epochs.

The detailed training algorithm of DMMV-A is summarized in Algorithm 1.

## C.3 Running Environment

The experiments are conducted on a Linux server (kernel 5.15.0-139) with 8x NVIDIA RTX 6000 Ada GPUs (48 GB each). The environment uses Python 3.12.8, PyTorch 2.5.1 with CUDA 12.4 and cuDNN 9.1. The key libraries include NumPy 2.1.3, Pandas 2.2.3, Matplotlib 3.10.0, SciPy 1.15.1, scikit-learn 1.6.1, and torchvision 0.20.1.

# D  More Experimental Results

## D.1  Comparison with All Baselines

Table 4 provides the full results of comparing DMMV-A and DMMV-S with all of the 14 baseline methods, which complements Table 1 in the paper. In Table 4, `Time-VLM`'s results on Illness dataset is marked by "–" since its paper doesn't report the results and its code is not publicly available at the time of this experiment. `CycleNet`'s paper doesn't report its results on Illness dataset, so we run its code and reproduce its results on Illness dataset in Table 4.

From Table 4, we can observe that DMMV-A maintains a clear advantage when compared against all of the baseline methods. It achieves 41 first-place results, significantly surpassing the second-best method. Additionally, taking a closer look at all compared methods, MMV-based methods LVM-based methods, and decomposition-based methods demonstrate superiority over other baseline methods. This suggests the synergy of MMV framework, LVMs, and decomposition strategy, which are explored by the proposed DMMV model.

## D.2  Further Analysis of The Inductive Bias

A contribution of our work lies in the in-depth analysis of an inductive bias of the current best LVM forecasters. In §3, we have discussed the impact of the alignment of the segment length and the period of time series on model performance. We find that the LVM exhibits a strong *inter-period consistency* when applied to synthetic data. The function of the synthetic time series is $x(t) = A(t) \cdot \sin\left(\frac{2\pi t}{P}\right)$, where the period $P$ is set to 24 and the amplitude function $A(t)$ decreases linearly over time. The forecasts are more accurate when the segment length is a multiple of the period (*e.g.*, 24, 48) than other values. This section provides detailed quantitative results on the synthetic data in Table 5. From Table 5, the fluctuations in MSEs and MAEs across different segment lengths other than 24 and 48 support the findings of the inductive bias toward "forecasting periods".

In addition, we evaluate the performance of the proposed method DMMV-A and `VisionTS` *w.r.t.* varying segment lengths to compare their robustness to the change of segment length. Fig. 10 summarizes the results in terms of MSE on four benchmark datsets, where the segment length varies from $\frac{P}{6}$ to $\frac{6P}{6}$ and $P$ is a period of the input time series. From Fig. 10, we have several observations. First, DMMV-A consistently outperforms `VisionTS`, validating the effectiveness of the proposed MMV framework. Second, in contrast to `VisionTS`, DMMV-A exhibits a better robustness to the change of segment length on ETTh1 and Weather datasets, but has a similar sensitivity to the change of segment length as `VisionTS` on ETTm1 and Illness datasets. This implies that by incorporating $f_{\text{num}}(\cdot)$, DMMV-A can alleviate $f_{\text{vis}}(\cdot)$'s sensitivity to the inductive bias to some extent. However, the current DMMV-A does not fully mitigate this limitation, suggesting a future work for method development as discussed in Appendix A.

Table 4: Full LTSF performance of the compared methods on the benchmark datasets. Lower MSE and MAE indicate better performance. The best performance is highlighted in **red**. Time-VLM results on the Illness dataset are unavailable in [52]. Its code was not publicly available at the time of this experiment. As such, its results on Illness dataset are marked by "−".

| Model | | DMMV-A | | DMMV-s | | Time-VLM | | VisionTS | | Time-LLM | | GPT4TS | | CALF | | CycleNet | | PatchTST | | TimesNet | | DLinear | | FEDformer | | Autoformer | | Stationary | | ETSformer | | Informer | |
|---|---|---|---|---|---|---|---|---|---|---|---|---|---|---|---|---|---|---|---|---|---|---|---|---|---|---|---|---|---|---|---|---|---|
| | Metric | MSE | MAE | MSE | MAE | MSE | MAE | MSE | MAE | MSE | MAE | MSE | MAE | MSE | MAE | MSE | MAE | MSE | MAE | MSE | MAE | MSE | MAE | MSE | MAE | MSE | MAE | MSE | MAE | MSE | MAE | MSE | MAE |
| ETTh1 | 96 | 0.354 | 0.389 | **0.350** | 0.388 | 0.361 | **0.386** | 0.355 | **0.386** | 0.376 | 0.402 | 0.370 | 0.389 | 0.370 | 0.393 | 0.374 | 0.396 | 0.370 | 0.399 | 0.384 | 0.402 | 0.375 | 0.399 | 0.376 | 0.419 | 0.449 | 0.459 | 0.513 | 0.491 | 0.494 | 0.479 | 0.865 | 0.713 |
| | 192 | **0.393** | **0.405** | 0.399 | 0.420 | 0.397 | 0.415 | 0.395 | 0.415 | 0.407 | 0.421 | 0.412 | 0.413 | 0.429 | 0.426 | 0.406 | 0.415 | 0.413 | 0.421 | 0.436 | 0.429 | 0.405 | 0.416 | 0.420 | 0.448 | 0.500 | 0.482 | 0.534 | 0.504 | 0.538 | 0.504 | 1.008 | 0.792 |
| | 336 | **0.387** | **0.413** | 0.399 | 0.415 | 0.420 | 0.421 | 0.419 | 0.421 | 0.430 | 0.438 | 0.448 | 0.431 | 0.451 | 0.440 | 0.431 | 0.430 | 0.422 | 0.436 | 0.491 | 0.469 | 0.439 | 0.416 | 0.459 | 0.465 | 0.521 | 0.496 | 0.588 | 0.535 | 0.574 | 0.521 | 1.107 | 0.809 |
| | 720 | 0.445 | 0.450 | 0.472 | 0.479 | **0.441** | 0.458 | 0.458 | 0.458 | 0.457 | 0.468 | **0.441** | **0.449** | 0.476 | 0.466 | 0.450 | 0.464 | 0.447 | 0.466 | 0.521 | 0.500 | 0.472 | 0.490 | 0.506 | 0.507 | 0.514 | 0.512 | 0.643 | 0.616 | 0.562 | 0.535 | 1.181 | 0.865 |
| | Avg. | **0.395** | **0.414** | 0.405 | 0.426 | 0.405 | 0.420 | 0.407 | 0.420 | 0.418 | 0.432 | 0.418 | 0.421 | 0.432 | 0.431 | 0.415 | 0.426 | 0.413 | 0.431 | 0.458 | 0.450 | 0.423 | 0.430 | 0.440 | 0.460 | 0.496 | 0.487 | 0.570 | 0.537 | 0.542 | 0.510 | 1.040 | 0.795 |
| ETTh2 | 96 | 0.294 | 0.349 | 0.286 | 0.360 | **0.267** | 0.335 | 0.288 | **0.334** | 0.286 | 0.346 | 0.280 | 0.335 | 0.284 | 0.336 | 0.279 | 0.341 | 0.274 | 0.336 | 0.340 | 0.374 | 0.289 | 0.353 | 0.358 | 0.397 | 0.346 | 0.388 | 0.476 | 0.458 | 0.340 | 0.391 | 3.755 | 1.525 |
| | 192 | 0.339 | 0.395 | 0.331 | 0.387 | **0.326** | **0.373** | 0.349 | 0.380 | 0.361 | 0.391 | 0.348 | 0.380 | 0.353 | 0.378 | 0.342 | 0.385 | 0.339 | 0.379 | 0.402 | 0.414 | 0.383 | 0.418 | 0.429 | 0.439 | 0.456 | 0.452 | 0.512 | 0.493 | 0.430 | 0.439 | 5.602 | 1.931 |
| | 336 | 0.322 | 0.384 | **0.309** | **0.378** | 0.357 | 0.406 | 0.364 | 0.398 | 0.390 | 0.414 | 0.380 | 0.405 | 0.361 | 0.394 | 0.371 | 0.413 | 0.329 | 0.380 | 0.452 | 0.452 | 0.448 | 0.465 | 0.496 | 0.487 | 0.482 | 0.486 | 0.552 | 0.551 | 0.485 | 0.479 | 4.721 | 1.835 |
| | 720 | 0.392 | 0.425 | 0.430 | 0.462 | 0.412 | 0.449 | 0.403 | 0.431 | 0.405 | 0.434 | 0.406 | 0.436 | 0.406 | 0.428 | 0.426 | 0.451 | **0.379** | **0.422** | 0.462 | 0.468 | 0.605 | 0.551 | 0.463 | 0.474 | 0.515 | 0.511 | 0.562 | 0.560 | 0.500 | 0.497 | 3.647 | 1.625 |
| | Avg. | 0.337 | 0.388 | 0.339 | 0.397 | 0.341 | 0.391 | 0.351 | 0.386 | 0.361 | 0.396 | 0.354 | 0.389 | 0.351 | 0.384 | 0.355 | 0.398 | **0.330** | **0.379** | 0.414 | 0.427 | 0.431 | 0.447 | 0.437 | 0.449 | 0.450 | 0.459 | 0.526 | 0.516 | 0.439 | 0.452 | 4.431 | 1.729 |
| ETTm1 | 96 | **0.279** | **0.329** | 0.296 | 0.349 | 0.304 | 0.346 | 0.284 | 0.332 | 0.291 | 0.341 | 0.300 | 0.340 | 0.323 | 0.350 | 0.299 | 0.348 | 0.290 | 0.342 | 0.338 | 0.375 | 0.299 | 0.343 | 0.379 | 0.419 | 0.505 | 0.475 | 0.386 | 0.398 | 0.375 | 0.398 | 0.672 | 0.571 |
| | 192 | **0.317** | **0.357** | 0.328 | 0.370 | 0.332 | 0.366 | 0.327 | 0.362 | 0.341 | 0.369 | 0.343 | 0.368 | 0.375 | 0.376 | 0.334 | 0.367 | 0.332 | 0.369 | 0.374 | 0.387 | 0.335 | 0.365 | 0.426 | 0.441 | 0.553 | 0.496 | 0.459 | 0.444 | 0.408 | 0.410 | 0.795 | 0.669 |
| | 336 | **0.351** | 0.381 | 0.369 | 0.393 | 0.364 | 0.383 | 0.354 | 0.382 | 0.359 | 0.379 | 0.376 | 0.386 | 0.411 | 0.401 | 0.368 | 0.386 | 0.366 | 0.392 | 0.410 | 0.411 | 0.369 | 0.386 | 0.445 | 0.459 | 0.621 | 0.537 | 0.495 | 0.464 | 0.435 | 0.428 | 1.212 | 0.871 |
| | 720 | 0.411 | 0.415 | **0.401** | 0.414 | 0.402 | **0.410** | 0.411 | 0.415 | 0.433 | 0.419 | 0.431 | 0.416 | 0.476 | 0.438 | 0.417 | 0.414 | 0.416 | 0.414 | 0.478 | 0.450 | 0.425 | 0.421 | 0.543 | 0.490 | 0.671 | 0.561 | 0.585 | 0.516 | 0.499 | 0.462 | 1.166 | 0.823 |
| | Avg. | **0.340** | **0.371** | 0.349 | 0.382 | 0.351 | 0.376 | 0.344 | 0.373 | 0.356 | 0.377 | 0.363 | 0.378 | 0.396 | 0.391 | 0.355 | 0.379 | 0.351 | 0.381 | 0.400 | 0.406 | 0.357 | 0.379 | 0.448 | 0.452 | 0.588 | 0.517 | 0.481 | 0.456 | 0.429 | 0.425 | 0.961 | 0.734 |
| ETTm2 | 96 | 0.172 | 0.260 | 0.164 | 0.254 | 0.160 | 0.250 | 0.174 | 0.262 | 0.162 | 0.248 | 0.163 | 0.249 | 0.177 | 0.255 | **0.159** | **0.247** | 0.165 | 0.255 | 0.187 | 0.267 | 0.167 | 0.260 | 0.203 | 0.287 | 0.255 | 0.339 | 0.192 | 0.274 | 0.189 | 0.280 | 0.365 | 0.453 |
| | 192 | 0.227 | 0.298 | 0.217 | 0.293 | 0.215 | 0.291 | 0.228 | 0.297 | 0.235 | 0.304 | 0.222 | 0.291 | 0.245 | 0.300 | **0.214** | **0.286** | 0.220 | 0.292 | 0.249 | 0.309 | 0.224 | 0.303 | 0.269 | 0.328 | 0.281 | 0.340 | 0.280 | 0.339 | 0.253 | 0.319 | 0.533 | 0.563 |
| | 336 | 0.272 | 0.327 | 0.273 | 0.332 | 0.270 | 0.325 | 0.281 | 0.337 | 0.280 | 0.329 | 0.273 | 0.327 | 0.309 | 0.341 | **0.269** | **0.322** | 0.274 | 0.329 | 0.321 | 0.351 | 0.281 | 0.342 | 0.325 | 0.366 | 0.339 | 0.372 | 0.334 | 0.361 | 0.314 | 0.357 | 1.363 | 0.887 |
| | 720 | 0.351 | 0.381 | 0.362 | 0.393 | **0.348** | 0.378 | 0.384 | 0.410 | 0.366 | 0.382 | 0.357 | **0.376** | 0.402 | 0.395 | 0.363 | 0.382 | 0.362 | 0.385 | 0.408 | 0.403 | 0.397 | 0.421 | 0.421 | 0.415 | 0.433 | 0.432 | 0.417 | 0.413 | 0.414 | 0.413 | 3.379 | 1.338 |
| | Avg. | 0.256 | 0.317 | 0.254 | 0.318 | **0.248** | 0.311 | 0.267 | 0.327 | 0.261 | 0.316 | 0.254 | 0.311 | 0.283 | 0.323 | 0.251 | **0.309** | 0.255 | 0.315 | 0.291 | 0.333 | 0.267 | 0.332 | 0.305 | 0.349 | 0.327 | 0.371 | 0.306 | 0.347 | 0.293 | 0.342 | 1.410 | 0.810 |
| Illness | 24 | 1.409 | **0.754** | 1.638 | 0.838 | − | − | 1.613 | 0.834 | 1.792 | 0.807 | 1.869 | 0.823 | 1.460 | 0.788 | 2.255 | 1.017 | **1.319** | **0.754** | 2.317 | 0.934 | 2.215 | 1.081 | 3.228 | 1.260 | 3.483 | 1.287 | 2.294 | 0.945 | 2.527 | 1.020 | 5.764 | 1.677 |
| | 36 | **1.290** | **0.745** | 1.323 | 0.753 | − | − | 1.316 | 0.750 | 1.833 | 0.833 | 1.853 | 0.854 | 1.573 | 0.837 | 2.121 | 0.950 | 1.430 | 0.834 | 1.972 | 0.920 | 1.963 | 0.963 | 2.679 | 1.080 | 3.103 | 1.148 | 1.825 | 0.848 | 2.615 | 1.007 | 4.755 | 1.467 |
| | 48 | **1.499** | **0.810** | 1.644 | 0.851 | − | − | 1.548 | 0.818 | 2.269 | 1.012 | 1.886 | 0.855 | 1.784 | 0.890 | 2.187 | 1.007 | 1.553 | 0.815 | 2.238 | 0.940 | 2.130 | 1.024 | 2.622 | 1.078 | 2.669 | 1.085 | 2.010 | 0.900 | 2.359 | 0.972 | 4.763 | 1.469 |
| | 60 | **1.428** | **0.773** | 1.473 | 0.810 | − | − | 1.450 | 0.783 | 2.177 | 0.925 | 1.877 | 0.877 | 1.982 | 0.962 | 2.185 | 0.997 | 1.470 | 0.788 | 2.027 | 0.928 | 2.368 | 1.096 | 2.857 | 1.157 | 2.770 | 1.125 | 2.178 | 0.963 | 2.487 | 1.016 | 5.264 | 1.564 |
| | Avg. | **1.407** | **0.771** | 1.520 | 0.813 | − | − | 1.482 | 0.796 | 2.018 | 0.894 | 1.871 | 0.852 | 1.700 | 0.869 | 2.187 | 0.992 | 1.443 | 0.798 | 2.139 | 0.931 | 2.169 | 1.041 | 2.847 | 1.144 | 3.006 | 1.161 | 2.077 | 0.914 | 2.497 | 1.004 | 5.137 | 1.544 |
| Electricity | 96 | **0.126** | **0.213** | 0.165 | 0.267 | 0.142 | 0.245 | 0.127 | 0.217 | 0.137 | 0.233 | 0.141 | 0.239 | 0.147 | 0.240 | 0.128 | 0.223 | 0.129 | 0.222 | 0.168 | 0.272 | 0.140 | 0.237 | 0.193 | 0.308 | 0.201 | 0.317 | 0.169 | 0.273 | 0.187 | 0.304 | 0.274 | 0.368 |
| | 192 | **0.145** | **0.237** | 0.172 | 0.276 | 0.157 | 0.260 | 0.148 | 0.237 | 0.152 | 0.247 | 0.158 | 0.253 | 0.163 | 0.254 | 0.144 | **0.237** | 0.157 | 0.240 | 0.184 | 0.289 | 0.153 | 0.249 | 0.201 | 0.315 | 0.222 | 0.334 | 0.182 | 0.286 | 0.199 | 0.315 | 0.296 | 0.386 |
| | 336 | 0.162 | 0.254 | 0.190 | 0.296 | 0.174 | 0.276 | 0.163 | 0.253 | 0.169 | 0.267 | 0.172 | 0.266 | 0.178 | 0.270 | **0.160** | 0.254 | 0.163 | 0.259 | 0.198 | 0.300 | 0.169 | 0.267 | 0.214 | 0.329 | 0.231 | 0.338 | 0.200 | 0.304 | 0.212 | 0.329 | 0.300 | 0.394 |
| | 720 | **0.197** | **0.286** | 0.242 | 0.344 | 0.214 | 0.308 | 0.199 | 0.293 | 0.200 | 0.290 | 0.207 | 0.293 | 0.215 | 0.300 | 0.198 | 0.287 | 0.197 | 0.290 | 0.220 | 0.320 | 0.203 | 0.301 | 0.246 | 0.355 | 0.254 | 0.361 | 0.222 | 0.321 | 0.233 | 0.345 | 0.373 | 0.439 |
| | Avg. | **0.158** | **0.248** | 0.192 | 0.296 | 0.172 | 0.272 | 0.159 | 0.250 | 0.165 | 0.259 | 0.170 | 0.263 | 0.176 | 0.266 | 0.158 | 0.250 | 0.162 | 0.253 | 0.193 | 0.295 | 0.166 | 0.264 | 0.214 | 0.327 | 0.227 | 0.338 | 0.193 | 0.296 | 0.208 | 0.323 | 0.311 | 0.397 |
| Weather | 96 | **0.143** | 0.195 | 0.168 | 0.218 | 0.148 | 0.200 | 0.146 | 0.200 | 0.155 | 0.199 | 0.148 | **0.188** | 0.168 | 0.207 | 0.167 | 0.221 | 0.149 | 0.198 | 0.172 | 0.220 | 0.176 | 0.237 | 0.217 | 0.296 | 0.266 | 0.336 | 0.173 | 0.223 | 0.197 | 0.281 | 0.300 | 0.384 |
| | 192 | **0.187** | 0.242 | 0.220 | 0.259 | 0.193 | 0.240 | 0.194 | 0.240 | 0.223 | 0.261 | 0.192 | **0.230** | 0.216 | 0.251 | 0.212 | 0.258 | 0.194 | 0.241 | 0.219 | 0.261 | 0.220 | 0.282 | 0.276 | 0.336 | 0.307 | 0.367 | 0.245 | 0.285 | 0.237 | 0.312 | 0.598 | 0.544 |
| | 336 | **0.237** | **0.273** | 0.267 | 0.304 | 0.243 | 0.281 | 0.243 | 0.275 | 0.251 | 0.279 | 0.246 | 0.273 | 0.271 | 0.292 | 0.260 | 0.293 | 0.245 | 0.282 | 0.280 | 0.306 | 0.265 | 0.319 | 0.339 | 0.380 | 0.359 | 0.395 | 0.321 | 0.338 | 0.298 | 0.353 | 0.578 | 0.523 |
| | 720 | **0.302** | 0.315 | 0.322 | 0.343 | 0.312 | 0.332 | 0.318 | 0.328 | 0.345 | 0.342 | 0.320 | 0.328 | 0.350 | 0.345 | 0.328 | 0.339 | 0.314 | 0.334 | 0.365 | 0.359 | 0.333 | 0.362 | 0.403 | 0.428 | 0.419 | 0.428 | 0.414 | 0.410 | 0.352 | 0.388 | 1.059 | 0.741 |
| | Avg. | **0.217** | 0.256 | 0.244 | 0.281 | 0.224 | 0.263 | 0.225 | 0.258 | 0.244 | 0.270 | 0.227 | **0.255** | 0.251 | 0.274 | 0.242 | 0.278 | 0.226 | 0.264 | 0.259 | 0.287 | 0.249 | 0.300 | 0.309 | 0.360 | 0.338 | 0.382 | 0.288 | 0.314 | 0.271 | 0.334 | 0.634 | 0.548 |
| Traffic | 96 | **0.344** | 0.237 | 0.362 | 0.253 | 0.393 | 0.290 | 0.346 | **0.232** | 0.392 | 0.267 | 0.396 | 0.264 | 0.416 | 0.274 | 0.397 | 0.278 | 0.360 | 0.249 | 0.593 | 0.321 | 0.410 | 0.282 | 0.587 | 0.366 | 0.613 | 0.388 | 0.612 | 0.338 | 0.607 | 0.392 | 0.719 | 0.391 |
| | 192 | **0.363** | 0.249 | 0.385 | 0.263 | 0.405 | 0.296 | 0.376 | **0.245** | 0.409 | 0.271 | 0.412 | 0.268 | 0.430 | 0.276 | 0.411 | 0.283 | 0.379 | 0.256 | 0.617 | 0.336 | 0.423 | 0.287 | 0.604 | 0.373 | 0.616 | 0.382 | 0.613 | 0.340 | 0.621 | 0.399 | 0.696 | 0.379 |
| | 336 | **0.387** | 0.256 | 0.396 | 0.262 | 0.420 | 0.305 | 0.389 | **0.252** | 0.434 | 0.296 | 0.421 | 0.273 | 0.451 | 0.286 | 0.424 | 0.289 | 0.392 | 0.264 | 0.629 | 0.336 | 0.436 | 0.296 | 0.621 | 0.383 | 0.622 | 0.337 | 0.618 | 0.328 | 0.622 | 0.396 | 0.777 | 0.420 |
| | 720 | 0.433 | **0.284** | 0.436 | 0.292 | 0.459 | 0.323 | **0.432** | 0.293 | 0.451 | 0.291 | 0.455 | 0.291 | 0.478 | 0.301 | 0.450 | 0.305 | 0.432 | 0.286 | 0.640 | 0.350 | 0.466 | 0.315 | 0.626 | 0.382 | 0.660 | 0.408 | 0.653 | 0.355 | 0.632 | 0.396 | 0.864 | 0.472 |
| | Avg. | **0.382** | 0.257 | 0.395 | 0.268 | 0.419 | 0.304 | 0.386 | **0.256** | 0.422 | 0.281 | 0.421 | 0.274 | 0.444 | 0.284 | 0.421 | 0.289 | 0.391 | 0.264 | 0.620 | 0.336 | 0.434 | 0.295 | 0.610 | 0.376 | 0.628 | 0.379 | 0.624 | 0.340 | 0.621 | 0.396 | 0.764 | 0.416 |
| # Wins | | **41** | | 4 | | 8 | | 9 | | 1 | | 7 | | 0 | | 11 | | 8 | | 0 | | 0 | | 0 | | 0 | | 0 | | 0 | | 0 | |

Table 5: Forecasting performance of an LVM *w.r.t.* varying segment length on a synthetic dataset. The function of the synthetic time series is $x(t) = A(t) \cdot \sin\left(\frac{2\pi t}{P}\right)$, where the period $P = 24$ and the amplitude function $A(t)$ decreases linearly over time.

| Segment Length | 16 | 20 | **24** | 28 | 32 | 36 | 40 | 44 | **48** |
|---|---|---|---|---|---|---|---|---|---|
| MSE | 0.043 | 0.099 | 0.001 | 0.147 | 0.154 | 0.143 | 0.221 | 0.114 | 0.002 |
| MAE | 0.177 | 0.257 | 0.024 | 0.342 | 0.347 | 0.315 | 0.408 | 0.289 | 0.045 |

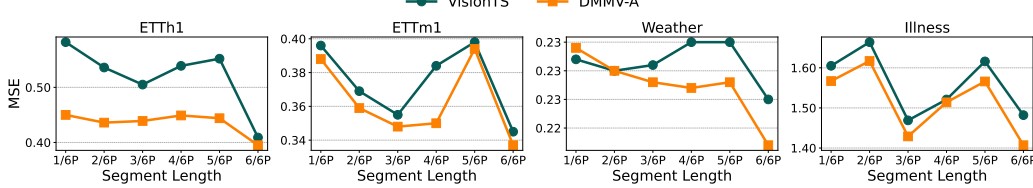

Figure 10: MSE Performance of DMMV-A and `VisionTS` *w.r.t.* varying segment length that is used in image construction. The $x$-axis indicates the segment length varies from $\frac{1}{6}$ period to $\frac{6}{6}$ period.

### D.3 Ablation Study

In Table 2 (§4.2), we provide ablation analyses for DMMV-A. Table 6 provides the ablation analysis for DMMV-S, where MSE and MAE are averaged over different prediction lengths. In addition, Tables 7 (Table 8) includes the full results for Table 2 (Table 6) with all prediction lengths.

In Table 6, from (a), replacing the linear numerical forecaster with `PatchTST` can slightly improve the performance of DMMV-S, likely because DMMV-S relies more on the predictions from the numerical view than visual view (Fig. 6). Therefore, in this case, increasing the complexity of the numerical model can improve the ability of $f_{num}(\cdot)$ and finally improve the overall performance. From (b), replacing `MAE` with `SimMiM` reduces the overall performance, this is the same as the findings in Table 2 for DMMV-A. From (c), gate-based fusion outperforms simple summation for DMMV-S, highlighting the effectiveness of gate fusion. From (d), fine-tuning the norm layers of $f_{vis}(\cdot)$ improves the performance for DMMV-S, suggesting the used fine-tuning strategy.

Table 6: Ablation analysis of DMMV-S. MSE and MAE are averaged over different prediction lengths. Lower MSE and MAE are better. "Improvement" of each ablation is relative to DMMV-S.

| Dataset ($\rightarrow$) | ETTh1 | | ETTm1 | | Illness | | Weather | |
|---|---|---|---|---|---|---|---|---|
| **Method ($\downarrow$), Metric ($\rightarrow$)** | MSE | MAE | MSE | MAE | MSE | MAE | MSE | MAE |
| DMMV-S | 0.405 | 0.426 | 0.349 | 0.382 | 1.520 | 0.813 | 0.244 | 0.281 |
| (a) $f_{num}(\cdot) \rightarrow$ `Transformer` | **0.402** | **0.423** | **0.342** | **0.376** | 1.544 | 0.841 | **0.229** | **0.264** |
| Improvement | 0.74% | 0.47% | 2.01% | 1.31% | -1.65% | -3.44% | 6.15% | 6.41% |
| (b) $f_{vis}(\cdot) \rightarrow$ `SimMIM` | 0.415 | **0.423** | 0.355 | 0.382 | 1.810 | 0.875 | 0.233 | 0.272 |
| Improvement | -2.47% | 0.47% | -1.72% | 0.00% | -19.16% | -7.63% | 4.92% | 3.20% |
| (c) Gate $\rightarrow$ Sum | 0.419 | 0.435 | 0.355 | 0.379 | **1.453** | **0.790** | 0.256 | 0.297 |
| Improvement | -3.46% | -2.24% | -2.01% | 0.52% | 4.34% | 2.95% | -4.51% | -5.69% |
| (d) Freeze $f_{vis}(\cdot)$ | 0.436 | 0.442 | 0.368 | 0.386 | 2.125 | 0.969 | 0.251 | 0.288 |
| Improvement | -7.41% | -3.76% | -5.75% | -1.05% | -39.83% | -19.07% | -2.46% | -2.14% |

### D.4 Additional Visualizations on Decomposition

Fig. 11 and Fig. 12 several more examples the decomposed time series of DMMV-S and DMMV-A. Fig. 11 illustrates a case where the series has a localized periodic anomaly at time step around 192, which poses a challenge for detecting periodic patterns. In this case, DMMV-A effectively suppresses the influence of the anomaly and extracts a clear periodic pattern from the time series series. In contrast, DMMV-S is affected by the anomaly and fails to capture a smooth periodic pattern. Fig. 12 is an example with weak periodicity, where the periodic signal is either faint or overwhelmed by trend. In this case, DMMV-A is able to extract and utilize the underlying periodicity to produce reasonable forecasts, which is better than DMMV-S, suggesting the importance of the

Table 7: Full results of the ablation analysis of DMMV-A. Lower MSE and MAE are better. The Illness dataset uses prediction lengths of $\{24, 36, 48, 60\}$ due to its short time series (in total 966 time steps), which is different from the prediction lengths of other datasets.

| Dataset($\rightarrow$) Method($\downarrow$), Metric($\rightarrow$) | Length | ETTh1 MSE | ETTh1 MAE | ETTm1 MSE | ETTm1 MAE | Illness MSE | Illness MAE | Weather MSE | Weather MAE |
|---|---|---|---|---|---|---|---|---|---|
| DMMV-A | 96 | 0.354 | 0.389 | 0.279 | 0.329 | 1.409 | 0.754 | 0.143 | 0.195 |
| | 192 | 0.393 | 0.405 | 0.317 | 0.357 | 1.290 | 0.745 | 0.187 | 0.242 |
| | 336 | 0.387 | 0.413 | 0.351 | 0.381 | 1.499 | 0.810 | 0.237 | 0.273 |
| | 720 | 0.445 | 0.450 | 0.411 | 0.415 | 1.428 | 0.773 | 0.302 | 0.315 |
| | Avg. | 0.395 | 0.414 | 0.340 | 0.371 | 1.407 | 0.771 | 0.217 | 0.256 |
| (a) $f_{num}(\cdot) \rightarrow$ Transformer | 96 | 0.357 | 0.389 | 0.279 | 0.329 | 1.604 | 0.823 | 0.145 | 0.193 |
| | 192 | 0.407 | 0.420 | 0.318 | 0.359 | 1.250 | 0.742 | 0.187 | 0.239 |
| | 336 | 0.389 | 0.411 | 0.352 | 0.382 | 1.555 | 0.803 | 0.241 | 0.283 |
| | 720 | 0.474 | 0.462 | 0.407 | 0.416 | 1.359 | 0.774 | 0.301 | 0.326 |
| | Avg. | 0.407 | 0.421 | 0.339 | 0.372 | 1.442 | 0.786 | 0.219 | 0.260 |
| (b) $f_{vis}(\cdot) \rightarrow$ SimMiM | 96 | 0.358 | 0.383 | 0.301 | 0.348 | 1.729 | 0.832 | 0.145 | 0.194 |
| | 192 | 0.405 | 0.41 | 0.325 | 0.363 | 1.643 | 0.734 | 0.192 | 0.242 |
| | 336 | 0.412 | 0.414 | 0.354 | 0.383 | 1.689 | 0.845 | 0.241 | 0.275 |
| | 720 | 0.453 | 0.452 | 0.398 | 0.412 | 1.534 | 0.845 | 0.328 | 0.332 |
| | Avg. | 0.407 | 0.415 | 0.345 | 0.377 | 1.649 | 0.814 | 0.227 | 0.261 |
| (c) Gate $\rightarrow$ Sum | 96 | 0.373 | 0.400 | 0.286 | 0.339 | 1.728 | 0.845 | 0.156 | 0.214 |
| | 192 | 0.414 | 0.424 | 0.329 | 0.369 | 1.423 | 0.795 | 0.204 | 0.261 |
| | 336 | 0.411 | 0.422 | 0.364 | 0.392 | 1.693 | 0.920 | 0.258 | 0.302 |
| | 720 | 0.457 | 0.461 | 0.427 | 0.430 | 1.580 | 0.890 | 0.315 | 0.335 |
| | Avg. | 0.414 | 0.427 | 0.352 | 0.383 | 1.606 | 0.863 | 0.233 | 0.278 |
| (d)BCMASK$\rightarrow$ No mask | 96 | 0.384 | 0.402 | 0.288 | 0.342 | 1.628 | 0.840 | 0.145 | 0.198 |
| | 192 | 0.413 | 0.440 | 0.325 | 0.363 | 1.325 | 0.796 | 0.191 | 0.244 |
| | 336 | 0.434 | 0.448 | 0.361 | 0.384 | 1.606 | 0.865 | 0.241 | 0.285 |
| | 720 | 0.474 | 0.473 | 0.421 | 0.419 | 1.414 | 0.811 | 0.308 | 0.340 |
| | Avg. | 0.426 | 0.441 | 0.349 | 0.377 | 1.493 | 0.828 | 0.221 | 0.267 |
| (e)BCMASK$\rightarrow$ Random mask | 96 | 0.348 | 0.384 | 0.279 | 0.329 | 1.618 | 0.859 | 0.146 | 0.197 |
| | 192 | 0.388 | 0.405 | 0.318 | 0.360 | 1.318 | 0.798 | 0.189 | 0.240 |
| | 336 | 0.383 | 0.404 | 0.350 | 0.381 | 1.560 | 0.858 | 0.243 | 0.282 |
| | 720 | 0.458 | 0.462 | 0.414 | 0.418 | 1.392 | 0.800 | 0.312 | 0.328 |
| | Avg. | 0.394 | 0.414 | 0.340 | 0.372 | 1.472 | 0.829 | 0.223 | 0.262 |
| (f) Freeze $f_{vis}(\cdot)$ | 96 | 0.389 | 0.402 | 0.293 | 0.342 | 1.482 | 0.761 | 0.161 | 0.224 |
| | 192 | 0.434 | 0.425 | 0.335 | 0.367 | 1.218 | 0.694 | 0.203 | 0.287 |
| | 336 | 0.431 | 0.428 | 0.372 | 0.389 | 1.58 | 0.82 | 0.285 | 0.302 |
| | 720 | 0.468 | 0.457 | 0.431 | 0.422 | 1.489 | 0.815 | 0.335 | 0.338 |
| | Avg. | 0.431 | 0.428 | 0.358 | 0.380 | 1.442 | 0.773 | 0.246 | 0.288 |
| (g) W/o decomposition | 96 | 0.352 | 0.387 | 0.274 | 0.329 | 1.728 | 0.938 | 0.143 | 0.195 |
| | 192 | 0.402 | 0.414 | 0.315 | 0.358 | 1.841 | 0.940 | 0.187 | 0.242 |
| | 336 | 0.391 | 0.410 | 0.347 | 0.382 | 1.672 | 0.886 | 0.237 | 0.284 |
| | 720 | 0.487 | 0.486 | 0.417 | 0.422 | 1.606 | 0.846 | 0.309 | 0.350 |
| | Avg. | 0.408 | 0.424 | 0.338 | 0.373 | 1.712 | 0.903 | 0.219 | 0.268 |

proposed adaptive decomposition method. In summary, the results demonstrate that DMMV-A has a strong modeling ability of temporal structures and robustness to fluctuations even when dealing with anomalous or weakly periodic time series, validating its reliability and applicability across a broad range of scenarios.

## D.5 Additional Visualizations on Masking Strategies

Fig. 13 and Fig. 14 present additional examples of BCMASK in DMMV-A. Similar to §4.3, Fig. 13 and Fig. 14 compare different masking methods. From both figures, we observe that BCMASK produces smooth patterns along the temporal ($x$-axis) dimension, effectively capturing periodic structures. Notably, when the input time series contains an anomaly (*e.g.*, Fig. 13, time steps 288-336), BCMASK can effectively extract the periodic patterns.

## D.6 Impact of Look-Back Window

Fig. 15 provides the MSE results that compare DMMV-A with the other three models. Fig. 15 demonstrate a similar trend as that of the MAE results in Fig. 9.

Table 8: Full results of the ablation analysis of DMMV-S. Lower MSE and MAE are better. The Illness dataset uses prediction lengths of $\{24, 36, 48, 60\}$ due to its short time series (in total 966 time steps), which is different from the prediction lengths of other datasets.

| Dataset(→)
Method(↓),  Metric(→) | Length | ETTh1 MSE | ETTh1 MAE | ETTm1 MSE | ETTm1 MAE | Illness MSE | Illness MAE | Weather MSE | Weather MAE |
|---|---|---|---|---|---|---|---|---|---|
| DMMV-S | 96 | 0.350 | 0.388 | 0.296 | 0.349 | 1.638 | 0.838 | 0.168 | 0.218 |
|  | 192 | 0.399 | 0.420 | 0.328 | 0.370 | 1.323 | 0.753 | 0.220 | 0.259 |
|  | 336 | 0.399 | 0.415 | 0.369 | 0.393 | 1.644 | 0.851 | 0.267 | 0.304 |
|  | 720 | 0.472 | 0.479 | 0.401 | 0.414 | 1.473 | 0.810 | 0.322 | 0.343 |
|  | Avg. | 0.405 | 0.426 | 0.349 | 0.382 | 1.520 | 0.813 | 0.244 | 0.281 |
| (a) $f_{\text{num}}(\cdot) \rightarrow$ Transformer | 96 | 0.352 | 0.387 | 0.286 | 0.339 | 1.613 | 0.829 | 0.148 | 0.194 |
|  | 192 | 0.401 | 0.420 | 0.325 | 0.364 | 1.417 | 0.825 | 0.193 | 0.240 |
|  | 336 | 0.395 | 0.415 | 0.354 | 0.387 | 1.610 | 0.853 | 0.246 | 0.280 |
|  | 720 | 0.460 | 0.471 | 0.401 | 0.414 | 1.536 | 0.858 | 0.330 | 0.341 |
|  | Avg. | 0.402 | 0.423 | 0.342 | 0.376 | 1.544 | 0.841 | 0.229 | 0.264 |
| (b) $f_{\text{vis}}(\cdot) \rightarrow$ SimMiM | 96 | 0.366 | 0.391 | 0.323 | 0.360 | 1.923 | 0.901 | 0.153 | 0.210 |
|  | 192 | 0.412 | 0.420 | 0.331 | 0.364 | 1.812 | 0.863 | 0.194 | 0.248 |
|  | 336 | 0.419 | 0.420 | 0.361 | 0.386 | 1.793 | 0.854 | 0.245 | 0.279 |
|  | 720 | 0.464 | 0.461 | 0.404 | 0.416 | 1.712 | 0.883 | 0.339 | 0.352 |
|  | Avg. | 0.415 | 0.423 | 0.355 | 0.382 | 1.810 | 0.875 | 0.233 | 0.272 |
| (c) Gate $\rightarrow$ Sum | 96 | 0.356 | 0.389 | 0.300 | 0.346 | 1.503 | 0.763 | 0.183 | 0.234 |
|  | 192 | 0.403 | 0.417 | 0.334 | 0.365 | 1.350 | 0.746 | 0.236 | 0.277 |
|  | 336 | 0.414 | 0.426 | 0.362 | 0.385 | 1.530 | 0.820 | 0.271 | 0.308 |
|  | 720 | 0.504 | 0.506 | 0.424 | 0.420 | 1.429 | 0.830 | 0.333 | 0.369 |
|  | Avg. | 0.419 | 0.435 | 0.355 | 0.379 | 1.453 | 0.790 | 0.256 | 0.297 |
| (d) Freeze $f_{\text{vis}}(\cdot)$ | 96 | 0.386 | 0.404 | 0.306 | 0.352 | 1.966 | 0.921 | 0.156 | 0.225 |
|  | 192 | 0.436 | 0.434 | 0.347 | 0.375 | 2.050 | 0.945 | 0.240 | 0.261 |
|  | 336 | 0.436 | 0.440 | 0.377 | 0.392 | 2.223 | 0.999 | 0.271 | 0.312 |
|  | 720 | 0.484 | 0.488 | 0.443 | 0.424 | 2.259 | 1.009 | 0.335 | 0.353 |
|  | Avg. | 0.436 | 0.442 | 0.368 | 0.386 | 2.125 | 0.969 | 0.251 | 0.288 |

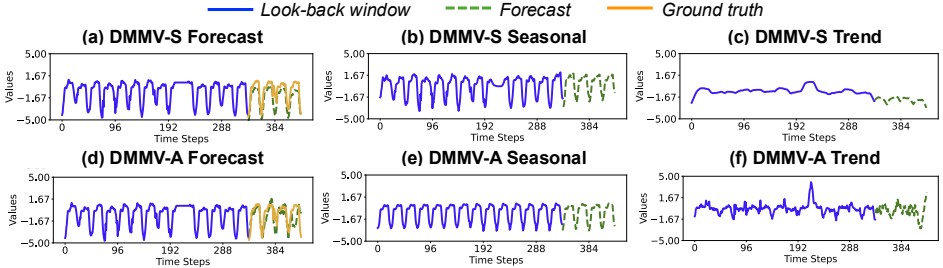

Figure 11: The decompositions of DMMV-S and DMMV-A on the same example in ETTh1: (a)(d) input time series and forecasts, (b)(e) seasonal component, and (c)(f) trend component.

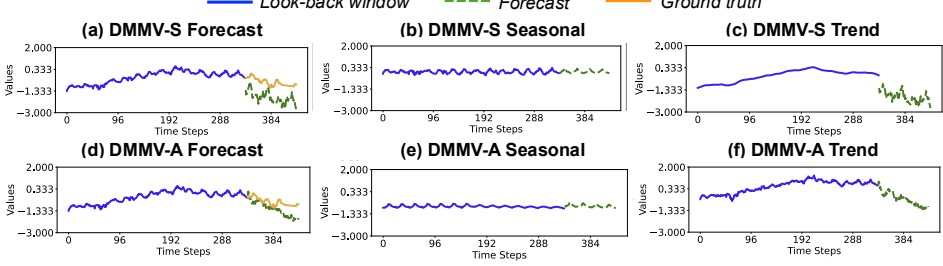

Figure 12: The decompositions of DMMV-S and DMMV-A on the same example in ETTh2: (a)(d) input time series and forecasts, (b)(e) seasonal component, and (c)(f) trend component.

## D.7  Standard Deviations

To assess the uncertainty and stability of the forecasting performance, we report the standard deviations of DMMV-S and DMMV-A on the four benchmark datasets used in §4.2 and §4.3 in Table 9. From Table 9, the relative standard deviations of the proposed models, which are calculated as

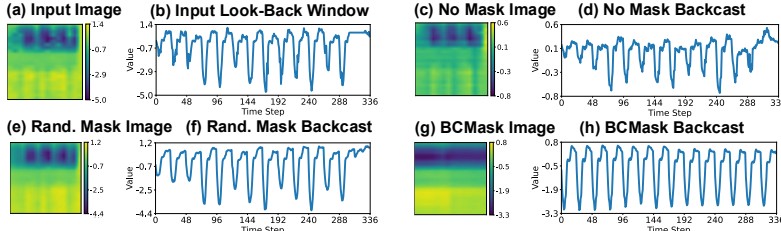

Figure 13: Comparison of different masking methods on the same example in ETTh1. (a) image of input look-back window; (c)(e)(g) are images of backcast output by DMMV-A: (c) uses "No mask"; (e) uses "Random mask"; (g) uses BCMASK. (b)(d)(f)(h) are their recovered time series, respectively.

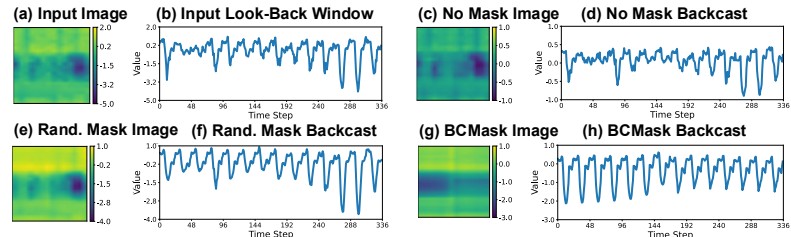

Figure 14: Comparison of different masking methods on the same example in ETTh2. (a) image of input look-back window; (c)(e)(g) are images of backcast output by DMMV-A: (c) uses "No mask"; (e) uses "Random mask"; (g) uses BCMASK. (b)(d)(f)(h) are their recovered time series, respectively.

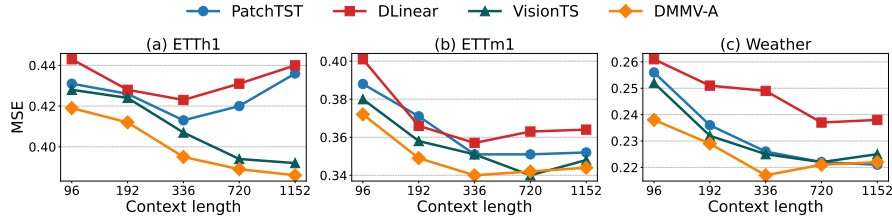

Figure 15: Average MSE comparison with varying look-back window (or context) lengths.

Table 9: Standard Deviations of DMMV-S and DMMV-A in terms of MSE and MAE on four LTSF benchmark datasets.

| Model | | DMMV-A | | DMMV-S | |
|---|---|---|---|---|---|
| Metric | | MSE | MAE | MSE | MAE |
| ETTh1 | 96 | 0.354± 0.001 | 0.390± 0.001 | 0.350 ± 0.001 | 0.388 ± 0.002 |
| | 192 | 0.393± 0.001 | 0.405± 0.001 | 0.399 ± 0.002 | 0.420 ± 0.001 |
| | 336 | 0.387± 0.001 | 0.413± 0.001 | 0.401 ± 0.002 | 0.415 ± 0.001 |
| | 720 | 0.447± 0.002 | 0.451± 0.001 | 0.472 ± 0.001 | 0.480 ± 0.002 |
| ETTm1 | 96 | 0.278± 0.001 | 0.329± 0.000 | 0.296 ± 0.001 | 0.348 ± 0.002 |
| | 192 | 0.317± 0.001 | 0.358± 0.001 | 0.328 ± 0.001 | 0.368 ± 0.002 |
| | 336 | 0.351± 0.001 | 0.381± 0.000 | 0.367 ± 0.002 | 0.393 ± 0.002 |
| | 720 | 0.411± 0.000 | 0.415± 0.000 | 0.401 ± 0.002 | 0.415 ± 0.003 |
| Illness | 24 | 1.409± 0.001 | 0.754± 0.001 | 1.638 ± 0.003 | 0.842 ± 0.005 |
| | 36 | 1.291± 0.002 | 0.742± 0.003 | 1.329 ± 0.012 | 0.751 ± 0.002 |
| | 48 | 1.499± 0.002 | 0.810± 0.011 | 1.643 ± 0.002 | 0.853 ± 0.005 |
| | 60 | 1.430± 0.003 | 0.774± 0.001 | 1.473 ± 0.002 | 0.810 ± 0.002 |
| Weather | 96 | 0.143± 0.001 | 0.196± 0.002 | 0.168 ± 0.001 | 0.218 ± 0.002 |
| | 192 | 0.187± 0.001 | 0.245± 0.003 | 0.221 ± 0.002 | 0.259 ± 0.002 |
| | 336 | 0.237± 0.001 | 0.272± 0.003 | 0.267 ± 0.002 | 0.305 ± 0.001 |
| | 720 | 0.300± 0.002 | 0.318± 0.003 | 0.323 ± 0.001 | 0.341 ± 0.003 |

the ratio between standard deviation and mean, are all below $1.30\%$ across different datasets and evaluation metrics, demonstrating their stability and robustness over different runs.

