# OpenReview forum: "Multi-Modal View Enhanced Large Vision Models for Long-Term Time Series Forecasting"
_NeurIPS.cc/2025/Conference — NeurIPS 2025 poster_

### Official Review · Reviewer_Ymur · 2025-06-29

**Clarity:** 2
**Significance:** 2
**Originality:** 2
**Rating:** 4
**Confidence:** 4

**Summary:**

The authors propose a decomposition-based long-term time series forecasting model where seasonal-like patterns are captured by a pretrained large vision model and trend signals are learned by a linear module. The components are merged using a lightweight gate function.

**Questions:**

Questions are listed above.

**Ethical Concerns:**

["NO or VERY MINOR ethics concerns only"]

**Final Justification:**

Based on the rebuttals, the authors addressed most concerns through statistical significance testing and technical clarifications. While performance gains are incremental, the work demonstrates some contributions through its adaptive decomposition framework and insights into periodicity bias in LVM-based time series forecasting.

**Limitations:**

As the authors acknowledge,  the triple usage of LVMs could cause some computational overhead, though they present it as the best solution available at the time of submission.

**Quality:**

2

**Strengths And Weaknesses:**

Strength:
1. The use of LVMs is well-motivated due to the structural similarity between continuous images and time series data.
2. They propose a backcast-residual mechanism to achieve more robust time series decomposition rather than relying on pre-defined components.

Weakness:
1. The claim that “applying LVMs to LTSF poses an inductive bias towards forecasting periods” is unclear. In my understanding, Fig. 3 shows that the LVMs do well when the segment length aligns with the true period of the time series. While I agree that preprocessing alignment in period-based transformation is important for LVMs, I am not convinced that this evidence supports the direct conclusion that LVMs are inherently good at seasonal modeling.
2. The sources of baseline results are inconsistent, and the authors do not explain why some baseline results differ from their original papers or whether they reproduced the methods themselves. It appears that the results for Time-LLM and GPT4TS are copied from VisionTS, while VisionTS results differ from the original paper. It would be better if the author could explain their selective use of results, as an inconsistent comparison setting may be unfair and may not reflect the performance gains.
3. Compared to the most competitive baseline VisionTS, the performance gains are marginal, with a maximum average gain is ~0.014. On ETTm1, Electricity, and Traffic datasets, the MSE improvements are less than 0.005. It could reflect that the multi-modal view of the TS plays a lesser role in forecasting than expected. I would suggest conducting a statistical significance test or providing the standard deviation for both methods to better understand the performance gap.
4. Another concern is about the cost-efficiency. The authors do not compare against the most competitive numerical methods like TimeMixer. As shown in their paper, TimeMixer with a larger window length could produce very strong results, which raises questions about whether LVMs are necessary in the decomposition-based setting. I am curious about the results if seasonal modeling is replaced by other numerical methods, like linear module used in TimeMixer. Would there still be significant gains from the usage of LVMs?

---

> ### Author Rebuttal · Authors · 2025-07-31
>
> Dear Reviewer Ymur,
>
> Thank you so much for the constructive feedback. We sincerely appreciate your valuable suggestions and questions. The following are our responses.
>
> **Q1: The claim that "applying LVMs to LTSF poses an inductive bias towards forecasting periods" is unclear. I am not convinced that LVMs are inherently good at seasonal modeling.**
>
> Thank you for pointing it out. By re-reading the draft, we became aware of this non-rigorous wording. Yes, we shouldn't say "LVMs poses an inductive bias towards forecasting periods", which may cause confusion since LVMs are not inherently designed for seasonal modeling. Instead, the bias comes from the period-based imaging method. We will clarify this difference in a revised version of the draft and emphasize the source of the bias is the period-based imaging method.
>
> **Q2: The sources of baseline results are inconsistent. It appears that the results for Time-LLM and GPT4TS are copied from VisionTS, while VisionTS results differ from the original paper. It would be better if the author could explain their selective use of results.**
>
> Yes, we'd like to clarify our criteria of reporting results. For Time-LLM and GPT4TS, we collected the results reproduced by the work in [*], which uses the standard evaluation protocol and re-implement the methods to confirm their reported results, adding confidence to the reliability of the results. Also, we have re-run both methods on selected datasets (ETT, Illness) and observed <1% differences, confirming consistency. For VisionTS, since it didn't use the standard evaluation protocol (its look-back window length varies, e.g., 1152 and 2304), we implemented it following the protocol (fixed look-back winodow as 336) for fair comparison. For CycleNet, we collected its results from its paper which uses the standard evaluation protocol. For other methods, we collected their results reproduced by [**] under the same protocol, which are generally better than those reported by their original papers. Since CycleNet didn't report results on Illness, we run it and report it below, which will be added to a revised version of the paper. Time-VLM didn't use Illness as well and released code recently. We will run it on Illness and add the results to the paper.
>
> CycleNet performance in Illness datasets:
>
> | Length | MSE   | MAE   |
> | ------ | ----- | ----- |
> | 24     | 2.255 | 1.017 |
> | 36     | 2.121 | 0.950 |
> | 48     | 2.187 | 1.007 |
> | 60     | 2.185 | 0.997 |
> | Avg.   | 2.187 | 0.992 |
>
> [*] M. Tan et al. "Are Language Models Actually Useful for Time Series Forecasting?". In NeurIPS, 2024.
> [**] M. Chen et al. "VisionTS: Visual Masked Autoencoders Are Free-Lunch Zero-Shot Time Series Forecasters". In ICML, 2025.
>
> **Q3: Compared to the most competitive baseline VisionTS, the performance gains are marginal. I would suggest conducting a statistical significance test or providing the standard deviation for both methods to better understand the performance gap.**
>
> Thank you for highlighting this concern. Yes, the challenge of improving MSE and MAE over the SOTA methods on the widely used benchmark datasets may raise doubts about the significance of improvements. To address this concern, we ran our method five times with different seeds and calculated standard deviations. In our appendix, Table 9 reports the standard deviations of DMMV on the same four datasets used for ablation and performance analysis (section 4.2, 4.3). In the following, we summarize the comparison between DMMV-A and the most competitive baseline, VisionTS.
>
> From the results, both methods show small standard deviations across multiple runs, suggesting their stability and the reliability of the results. We ran two-sample t-tests for each dataset under different forecasting horizons (i.e., 96, 192, 336, 720). DMMV-A exhibits significant improvements over VisionTS in 13 out of 16 cases using MSE (p-value<0.05) and 9 out of 16 cases using MAE (p-value<0.05), totaling a majority of the cases. Also, considering the non-trivial difference in the overall number of wins between DMMV-A and VisionTS in Table 1, i.e., 43 and 9, we think the improvements are not a matter of randomness. We will extend this analysis to all datasets and add the results to a revised version of the appendix.
>
> | Datasets | Predicted length | DMMV-A(MSE) | DMMV-A(MAE) | VisionTS(MSE) | VisionTS(MAE) |
> | --- | --- | --- | --- | --- | --- |
> | ETTh1 | 96 | 0.354$\pm$0.001 | 0.390$\pm$0.001 | 0.355$\pm$0.001 | 0.386$\pm$0.001 |
> |  | 192 | 0.393$\pm$0.001 | 0.405$\pm$0.001 | 0.395$\pm$0.003 | 0.407$\pm$0.000 |
> |  | 336 | 0.387$\pm$0.001 | 0.413$\pm$0.001 | 0.419$\pm$0.000 | 0.421$\pm$0.000 |
> |  | 720 | 0.447$\pm$0.002 | 0.451$\pm$0.001 | 0.458$\pm$0.002 | 0.460$\pm$0.001 |
> | ETTm1 | 96 | 0.278$\pm$0.001 | 0.329$\pm$0.000 | 0.284$\pm$0.001 | 0.332$\pm$0.001 |
> |  | 192 | 0.317$\pm$0.001 | 0.358$\pm$0.001 | 0.327$\pm$0.001 | 0.362$\pm$0.001 |
> |  | 336 | 0.351$\pm$0.001 | 0.381$\pm$0.000 | 0.354$\pm$0.001 | 0.382$\pm$0.001 |
> |  | 720 | 0.411$\pm$0.000 | 0.415$\pm$0.000 | 0.411$\pm$0.001 | 0.415$\pm$0.000 |
> | illness | 24 | 1.409$\pm$0.001 | 0.754$\pm$0.001 | 1.613$\pm$0.001 | 0.834$\pm$0.000 |
> |  | 36 | 1.291$\pm$0.002 | 0.742$\pm$0.003 | 1.316$\pm$0.002 | 0.750$\pm$0.001 |
> |  | 48 | 1.499$\pm$0.002 | 0.810$\pm$0.011 | 1.548$\pm$0.002 | 0.818$\pm$0.001 |
> |  | 60 | 1.430$\pm$0.003 | 0.774$\pm$0.001 | 1.450$\pm$0.001 | 0.783$\pm$0.001 |
> | Weather | 96 | 0.143$\pm$0.001 | 0.196$\pm$0.002 | 0.146$\pm$0.000 | 0.191$\pm$0.000 |
> |  | 192 | 0.187$\pm$0.001 | 0.245$\pm$0.003 | 0.194$\pm$0.001 | 0.238$\pm$0.001 |
> |  | 336 | 0.237$\pm$0.001 | 0.272$\pm$0.003 | 0.243$\pm$0.000 | 0.275$\pm$0.000 |
> |  | 720 | 0.300$\pm$0.002 | 0.318$\pm$0.003 | 0.318$\pm$0.004 | 0.328$\pm$0.001 |
>
> Additionally, we'd like to highlight another advantage of introducing TS view by our method: It alleviates the sensitivity to period estimation. When the period $P$ is perturbed (i.e., varying by P/6, 2P/6, ..., P), VisionTS's performance drops more (e.g., ETTh1: 0.409 → 0.582 (42.3%)) than our model (e.g., ETTh1: 0.395 → 0.450 (13.9%)), suggesting the better robustness of our method.
>
> **Q4: I am curious about the results if seasonal modeling is replaced by other numerical methods, like linear module used in TimeMixer. Would there still be significant gains from the usage of LVMs?**
>
> This is an interesting question, which can help validate the effectiveness of LVMs in our framework. We performed an ablation by replacing the LVM in DMMV-A (Figure 1(b)) with the seasonal mixing module of TimeMixer [*]. Specifically, we use this module to extract seasonal components from the look-back window and use DMMV-A's numerical forecaster to model the residual component. The forecasts from the numerical forecaster and the seaonal mixing module are aggrated by the gate function. The ablation results are summarized as following. As can be seen, the original DMMV-A outperforms the variants, suggesting the effectiveness of visual forecasters.
>
> | Dataset | DMMV-A (MSE) | DMMV-A (MAE) | Seasonal mixing (MSE)     | Seasonal mixing     (MAE) |
> |---------|-------------|---------------|---------------------------|---------------------------|
> | ETTh1   | 0.395       | 0.414         | 0.436                     | 0.448                     |
> | ETTm1   | 0.340       | 0.371         | 0.365                     | 0.389                     |
> | Illness | 1.407       | 0.771         | 2.378                     | 1.105                     |
> | Weather | 0.217       | 0.256         | 0.250                     | 0.306                     |
>
> [*] S. Wang et al. "TimeMixer: Decomposable Multiscale Mixing for Time Series Forecasting". In ICLR, 2024.
>
> **Q5 in Limitations: As the authors acknowledge, the triple usage of LVMs could cause some computational overhead, though they present it as the best solution available at the time of submission.**
>
> Yes, we acknowledged this limitation in our appendix A. The triple usage includes two passes of LVMs for backcasting and one pass for forecasting. At the time of submission, we thought two passes were possibily the minimum usage of LVM for backcasting. Compared to other LVM-based models, which include at least one pass for forecasting, our method adds two backcast passes. In the following, we compared the inference time of our method and VisionTS. From the results, the computational overhead of DMMV-A looks practically acceptable. We will add these results and the discussion to a revised version of the paper. Moreover, we'd like to improve the model by optimizing the backcast-forecast mechanism using LVMs. Some thoughts are provided in appendix A.
>
> The Averaged inference time for each sample in ETTh1:
> | Model | Inference time  |
> | --- | --- |
> | DMMV-A | 0.213 ms |
> | VisionTS | 0.165 ms|

---

> > ### Comment · Reviewer_Ymur · 2025-08-04
> >
> > Thank you for the responses! Though my concern about the marginal improvements remains, I will consider the perspectives of other reviewers in my final assessment.

---

> > > ### Author Response · Authors · 2025-08-08
> > > **Response to Reviewer Ymur**
> > >
> > > Dear Reviewer Ymur,
> > >
> > > As it is approaching to the end of the discussion period, we would be very grateful if we could hear from you to see what might be missing in our previous response to Q3 that makes the concern about the improvements remain. We'd like to take the opportunity before the end of the discussion period to address any further questions. Thank you for your attention!

---

> > > > ### Comment · Reviewer_Ymur · 2025-08-09
> > > >
> > > > Thank you for the follow-up messages. I acknowledge that the two-sample t-tests with n=5 show statistical differences between the proposed method and VisionTS in the majority of cases. However, my primary concern remains about the practical significance. The absolute performance gains remain small in some cases. Despite this weakness, I agree that the paper provides some insights into the periodicity bias and adaptive decomposition mechanism. I will reconsider my rating.

---

> ### Author Response · Authors · 2025-08-07
> **Response to Reviewer Ymur**
>
> Dear Reviewer Ymur,
>
> Thank you for reviewing our response!
>
> We humbly think the statistical tests with p-values in our previous response to Q3 are sufficient to validate that the improvements are not marginal.
>
> Could you please let us know what might be missing in our previous response to Q3 so that we can take the opportunity to address the remaining concerns?
>
> We hope by clarifying the concerns our work can meet the reviewer's standards for a positive rating. Thank you for your attention.

---

> ### Author Response · Authors · 2025-08-09
> **Thank you for recognizing this work's merits!**
>
> Dear Reviewer Ymur,
>
> Thank you for the further response! We agree with the reviewer that the proposed model is not a panacea for each case and each dataset, like many SOTA models in this domain. However, we humbly think the overall improvement that the proposed method exhibits in the majority of the evaluation cases (across datasets, forecasting horizons, and evaluation metrics) is a reasonable indicator of non-trivial advancement, which we believe the reviewer also values.
>
> We sincerely appreciate the reviewer's recognition of this work's merits, including the finding on the bias of the SOTA LVM-based forecasters and the effective solution to address the limitation by a backcast-residual based adaptive decomposition framework, which may inspire further developments. We also thank the reviewer for reconsidering the rating based on our discussion.

---

### Official Review · Reviewer_KH6N · 2025-06-30

**Clarity:** 3
**Significance:** 3
**Originality:** 3
**Rating:** 4
**Confidence:** 5

**Summary:**

The authors propose an approach for long-horizon forecasting using large vision models. In their approach, the authors use a pre-trained large vision model to model the periodic components of a time series, and another architecture to model its trend components. The authors show that large vision models have inductive biases which allow them to model periodic components well. Their approach outperforms compared long-horizon forecasting models on a widely-used benchmark.

**Questions:**

I do not have any other questions beyond the ones I have asked in the previous section.

**Ethical Concerns:**

["NO or VERY MINOR ethics concerns only"]

**Final Justification:**

I have read the authors' rebuttal and have responded to them.

I believe that my current score reflects my assessment of the paper in its current form. The datasets need to be improved and the authors are running experiments to this end. I also do not immediately see the practical value of the method, with the baselines that the model is compared against. That said I do think it is an interesting direction of work.

**Limitations:**

The authors have a section on the limitations of their in the appendix. I would encourage the authors to reflect on the efficiency of your method.

**Paper Formatting Concerns:**

I do not have any concerns with the formatting of the paper.

**Quality:**

3

**Strengths And Weaknesses:**

## Strengths
1. The approach is interesting, and performs well in comparison to the compared methods.
2. The paper is well-written and fairly easy to follow.
3. The related work is well written and organized.

## Weaknesses
1. **Comparison on the GIFT-Eval benchmark:** The datasets that the authors have used to evaluate their methods have know issues. The GIFT-Eval forecasting benchmark was released to fix some of these issues. I would encourage the authors to compare their model on the GIFT-Eval benchmark.
2. **SoTA compared methods:** The model is not compared against SoTA forecasting methods such as N-HITs, iTransformer etc., and foundation models such as TimesFM, MOMENT, MOIRAI, Chronos, LagLlama, etc.
3. **Efficiency:** The method is not efficient, and there should be research on
4. **Value of multimodality:** The value of multimodality is non immediately clear to me. How does the model do without explicitly modeling the linear components (only using the vision model), versus only using a transformer / linear component, versus model decomposed time series with a linear model (DLinear) or a transformer (say decomposition + PatchTST)?
5. Why is 336 chosen as the look back window? In most recent studies, lookback windows are either 512 (e.g., MOMENT, PatchTST), or 96 (e.g., iTransformer).

---

> ### Author Rebuttal · Authors · 2025-07-31
>
> Dear Reviewer KH6N,
>
> Thank you so much for the constructive feedback. We sincerely appreciate your valuable suggestions and questions. The following are our responses.
>
> **Q1: Comparison on the GIFT-Eval benchmark: I would encourage the authors to compare their model on the GIFT-Eval benchmark.**
>
> Thank you for this valuable suggestion. We acknowledge the limitations of the widely used eight benchmark datasets. We used them mainly for their widespread adoption in related works, so as to ensure fair comparison and reproducibility. While our method is not intended as a large-scale foundation model and was therefore not initially pre-trained on datasets like GIFT-Eval or Time-MMD, we fully agree that evaluating on standardized and diverse benchmarks such as GIFT-Eval is helpful for demonstrating generalization and robustness.
>
> Due to time constraints, we have currently finished running the solar dataset from GIFT-Eval. We present the performance of DMMV-A and compare it with VisionTS on this dataset in the table below, which demonstrates the superiority of DMMV-A. We'd like to extend the results to more GIFT-Eval datasets and include them in a revised version of the paper.
>
> | model        | DMMV-A |      |      |      | VisionTS |      |      |      |
> | ------------ | ------ | ---- | ---- | ---- | -------- | ---- | ---- | ---- |
> | Input Length | MSE    | MAE  | MAPE | CRPS | MSE      | MAE  | MAPE | CRPS |
> | 96           | 0.187  | 0.243| 1.996| 0.415| 0.190    | 0.245| 2.081| 0.475|
> | 192          | 0.201  | 0.252| 2.013| 0.434| 0.206    | 0.257| 2.165| 0.492|
> | 336          | 0.205  | 0.253| 2.158| 0.508| 0.214    | 0.265| 2.231| 0.512|
> | 720          | 0.210  | 0.257| 2.162| 0.512| 0.235    | 0.299| 2.342| 0.532|
> | Avg.         | 0.201  | 0.251| 2.082| 0.467| 0.211    | 0.267| 2.204| 0.502|
>
>
> **Q2: SoTA compared methods: The model is not compared against SoTA forecasting methods such as N-HITs, iTransformer, etc., and foundation models such as TimesFM, MOMENT, MOIRAI, Chronos, LagLlama, etc.**
>
> Thank you for raising this concern. Our baseline selection aimed to cover diverse models by categories (Table 4 in appendix): (1) multimodal (Time-VLM), (2) vision-based (VisionTS), (3) language-based (GPT4TS, Time-LLM, CALF), (4) numerical methods (PatchTST, CycleNet, FEDformer, etc.).
>
> Models like **N-HiTS** and **iTransformer** were not included because they belong to numerical methods similar to baselines reported in that category (e.g., PatchTST), which empirically perform no better than the cross-modal baselines[*]. Time series foundation models (TimesFM, MOMENT, Moirai, Chronos, Lag-Llama) rely on pre-training on large-scale time series datasets, which differs fundamentally from our experimental setting that didn't use time series pre-training data, which may lead to unfair comparisons.
>
> Averaged Results for N-HiTS
> | Dataset | MSE | MAE |
> | --- | --- | --- |
> | ETTh1 | 0.473 | 0.462 |
> | ETTm1 | 0.395 | 0.412 |
> | illness | 2.997 | 1.171 |
> | Weather | 0.235 | 0.286 |
>
> [*] M. Jin et al. "Time-LLM: Time Series Forecasting by Reprogramming Large Language Models". In ICLR, 2024.
>
> **Q3: Efficiency: The method is not efficient, and there should be research on.**
>
> We agee with the reviewer that our method is not designed to be the most efficient but rather to balance performance and complexity. The additional numerical forecaster in DMMV-A is lightweight (0.2% of total weights) and adds only a small number of parameters compared to the LVM backbone. The table below compares the total parameters and training time per epoch on the ETTh1 dataset.
>
> In practice, our method only incurs a negligible overhead compared to VisionTS, making the it suited for real-world applications while still delivering significant performance gains, especially when dealing with non-periodic datasets or cases where the predefined period P is inaccurate. Consider the benefits, we think the small cost overhead is paid off. We'd like to include this discussion in the revised draft.
>
> | Model | Parameters | Training Time / Epoch |
> | --- | --- | --- |
> | DMMV-A | 112.1M | 158s |
> | VisionTS | 111.9M | 133s |
>
> **Q4: Value of multimodality: How does the model do without explicitly modeling the linear components (only using the vision model), versus only using a transformer / linear component, versus model decomposed time series with a linear model (DLinear) or a transformer (say decomposition + PatchTST)?**
>
> Yes, we agree with the reviewer that comparing with the suggested ablations will help understand the advantage of multimodal views.
>
> For "model only using the vision model", the VisionTS baseline in Table 1 serves for this purpose since DMMV-A's visual forecater alone is similar to VisionTS. For "model only using a transformer / linear component", the PatchTST and DLinear in Table 1 serve for this purpose since DMMV-A's numerical forecaster alone is similar to either of them. However, please let us clarify that DMMV-A's technical novelty lies in the multimodal view fusion framework, i.e., the backcast-residual mechanism in Figure 1(b), by leveraging the inductive bias of the visual forecaster.
>
> From Table 1, DMMV-A outperforms the aforementioned baselines, suggesting multimodal fusion brings complementary advantages.
>
> For "model decomposed time series with a linear model or a transformer", we replaced the visual forecaster in DMMV-A with a linear model, and report the results in the following. As can be seen, the original DMMV-A outperforms the variants, suggesting the effectiveness of visual forecasters.
>
> | Dataset | DMMV-A (MSE) | DMMV-A (MAE) | DMMV-Linear (MSE)     | DMMV-Linear     (MAE) |
> |---------|-------------|---------------|---------------------------|---------------------------|
> | ETTh1   | 0.395       | 0.414         | 0.436                     | 0.448                     |
> | ETTm1   | 0.340       | 0.371         | 0.365                     | 0.389                     |
> | Illness | 1.407       | 0.771         | 2.378                     | 1.105                     |
> | Weather | 0.217       | 0.256         | 0.250                     | 0.306                     |
>
>
> **Q5: Why is 336 chosen as the look-back window? In most recent studies, lookback windows are either 512 (e.g., MOMENT, PatchTST), or 96 (e.g., iTransformer).**
>
> We appreciate the opportunity to clarify this point. While 512 and 96 are commonly used look-back lengths, 336 is also a standard choice adopted by models such as PatchTST (PatchTST/42 version), DLinear, CycleNet, ETSformer. The key reason for adopting 336 is that we tried to follow the standard evaluation protocol of our baselines as specified in [1][2][3]. This facilitates fair comparisons and reproducibility.
>
> To validate robustness, we conducted experiments with multiple window sizes (Illness: {104, 156, 208, 312, 416}). The averaged MSE results in the table below indicates DMMV-A outperforms the baseline in most cases. Also, in Figure 9 of the paper and Figure 15 of the appendix, our method consistently outperforms baselines under various input windows, confirming its robustness to look-back length.
>
> Models Averaged performance under different look-back windows:
>
> | Model | Dataset | 96 | 192 | 336 | 720 | 1152 |
> | --- | --- | --- | --- | --- | --- | --- |
> | VisionTS | ETTh1 | 0.428 | 0.424 | 0.407 | 0.394 | 0.392 |
> |  | ETTm1 | 0.380 | 0.358 | 0.351 | 0.340 | 0.348 |
> |  | weather | 0.252 | 0.232 | 0.225 | 0.222 | 0.225 |
> |  | Illness | 1.482 | 1.429 | 1.523 | 1.729 | 2.112 |
> | DMMV-A | ETTh1 | 0.419 | 0.412 | 0.395 | 0.389 | 0.386 |
> |  | ETTm1 | 0.372 | 0.349 | 0.340 | 0.342 | 0.344 |
> |  | weather | 0.238 | 0.229 | 0.217 | 0.221 | 0.222 |
> |  | Illness | 1.407 | 1.389 | 1.497 | 1.575 | 1.494 |
>
>
> [1] M. Tan et al. "Are Language Models Actually Useful for Time Series Forecasting?". In NeurIPS, 2024.
>
> [2] A. Zeng et al. "Are Transformers Effective for Time Series Forecasting?". In AAAI, 2023.
>
> [3] S. Lin et al. "CycleNet: Enhancing Time Series Forecasting through Modeling Periodic Patterns". In NeurIPS, 2024.
>
> **Q6 in Limitations:  The authors have a section on the limitations of their in the appendix. I would encourage the authors to reflect on the efficiency of your method.**
>
> This question is related to Q3. The response is summarized in the reponse to Q3. We'd like to follow the suggestion and include it the discussion of limitation in Appendix A.

---

> > ### Comment · Reviewer_KH6N · 2025-08-03
> > **Comments on the rebuttal**
> >
> > Dear Authors,
> >
> > Thank you so much for your work!
> >
> > Thank you so much for starting to run experiments on the GIFT Eval framework. I believe experiments on this benchmark would be important to understand the merits of the proposed method.
> >
> > > We used them mainly for their widespread adoption in related works, so as to ensure fair comparison and reproducibility
> >
> > Experiments on this benchmark does give the reader a signal, albeit a noisy one. Improving the benchmark (by evaluating the model on GIFT Eval) does *not* make the comparison unfair or non-reproducible.
> >
> > > Models like N-HiTS and iTransformer were not included because they belong to numerical methods similar to baselines reported in that category (e.g., PatchTST), which empirically perform no better than the cross-modal baselines[*].
> >
> > TimeLLM is evaluated on the same datasets with known issues. Therefore, I humbly disagree with this claim.
> >
> > Besides, it is unclear to me what the goal of the paper is. Is it to propose a method that is practical, or just interesting? If it is the latter then I agree that signals from some datasets and a limited number of baselines (including VisionTS) might be _enough_. But if the goal of the method is to be practical, in that case, its efficiency and performance must be compared against state-of-the-art models, including numerical only foundation models.
> >
> > I would like to keep my score.

---

> > > ### Author Response · Authors · 2025-08-09
> > >
> > > Dear Reviewer KH6N,
> > >
> > > Thanks again for reviewing our response! We hope our response to your remaining questions helps address the remaining concerns. If not, we'd like to take the opportunity before the end of the discussion period to address any further questions. Thank you!

---

> ### Author Response · Authors · 2025-08-05
> **Thank you for reviewing our response!**
>
> Dear Reviewer KH6N,
>
> Thank you for reviewing our response! We appreciate your recognition of our new experiments on the GIFT Eval framework. We will complete these experiments to ensure comprehensive comparisons. Meanwhile, please let us address your remaining concerns in the following.
>
> **1. Improving the benchmark (by evaluating the model on GIFT Eval) does not make the comparison unfair or non-reproducible.**
>
> We totally agree with this statement. In our previous response to Q1, we stated "We used them (i.e., the 8 datasets) mainly for their widespread adoption in related works, so as to ensure fair comparison and reproducibility" because we wanted to provide the reason for our earlier focus on the 8 benchmark datasets used in our experiments. Yes, adding GIFT Eval to the 8 datasets will enhance the experiments.
>
> **2. TimeLLM is evaluated on the same datasets with known issues. Therefore, I humbly disagree with this claim.**
>
> Thank you for pointing this out. Considering the importance of N-HiTS and iTransformer, we compared them with our model on three datasets given the time limitation. The results in the following table validates DMMV-A's effectiveness in leveraging multimodal views to improve performance over N-HiTS and iTransformer. We'd like to extend the experiments to all datasets, including GIFT Eval, and add them to our current 14 baselines in Table 4 in the appendix.
>
> | **Datasets**  | **DMMV-A** | **iTransformer** | **N-HiTS** |
> | ----------- | ---------- | ---------------- | ---------- |
> | Electricity | 0.158      | 0.178            | 0.185      |
> | Traffic     | 0.382      | 0.428            | 0.452      |
> | Weather     | 0.217      | 0.258            | 0.248      |
>
> **3. Besides, it is unclear to me what the goal of the paper is. Is it to propose a method that is practical, or just interesting?**
>
> We apologize if our purpose causes any confusion. Please let us clarify it. The goal of the paper includes (1) highlighting a finding about the SOTA LVM-based forecasters: bias towards forecasting periods caused by imaging technique; (2) proposing an effective solution to address the limitation by a backcast-residual based multimodal view framework, so as to inspire further developments; (3) providing a practically useful model.
>
> We think the reviewer focuses on the third point, which we value a lot. To this end, we'd like to complete our experiments on GIFT Eval and add N-HiTS and iTransformer to our current 14 SOTA baselines (including models released in 2024 and 2025) in Table 4 in the appendix. As for time series foundation models (TSFMs), we humbly think comparing a model (e.g., DMMV-A) leveraging **cross-modal knowledge** (e.g., visual knowledge) with a TSFM leveraging **intra-modal knowledge** (i.e., obtained by pre-training on large-scale time series data) is unfair to some extent since the sources of knowledge are different. However, we think it is promising to combine vision models and TSFMs for fusing cross-modal knowledge and intra-modal knowledge due to their complementary modalities, which may further enhance time series modeling.
>
> In addition to quantitative comparsions, we think analyses to support the first and second points are worth attention, which corresponds to our evaluations in Section 4.3.
>
> As for efficiency, in addition our previous response to Q3, we'd like to provide the average inference time of our method DMMV-A, using VisionTS as a reference point, in the following. Since DMMV-A only adds a light-weight numerical forecaster to the visual forecaster, the inference overhead is small and the overall inference time looks practically reasonable.
>
> As such, we hope the full perspectives of our goal, including the intellectual merits and the practical solution, could be fairly assessed. Meanwhile, we'd appreciate it if the reviewer could let us know what else might be missing so that we can take the opportunity to clarify it. Thank you!
>
> The Averaged inference time for each sample in ETTh1:
> | Model | Inference time  |
> | --- | --- |
> | DMMV-A | 0.213 ms |
> | VisionTS | 0.165 ms|

---

### Official Review · Reviewer_srxJ · 2025-07-02

**Clarity:** 3
**Significance:** 2
**Originality:** 3
**Rating:** 3
**Confidence:** 4

**Summary:**

The paper proposes a multi-model view framework (DMMV) that harnesses the inductive bias of LVM-based forecaster and the strength of numerical forecaster. Extensive experiments show that DMMV outperforms single-view and multi-modal view baselines, achieving good prediction results.

**Questions:**

1. For time series data with clear periodicity, DMMV leverages the inductive bias of vision models to capture such patterns effectively. However, this makes the model highly sensitive to the choice of segment length P; if P is not properly set, the performance may deteriorate significantly.

2. Compared to DMMV-A, DMMV-S incorporates more complex designs, such as the backcast-residual mechanism. However, based on the results in Figure 5, DMMV-A outperforms DMMV-S. Does this suggest that these additional design components may be ineffective or unnecessary?

3. Table 1 does not report the results for DMMV-S. Could the authors include them for completeness and comparison?

4. On datasets with strong periodicity, such as Electricity and Traffic, the performance gains of DMMV-A over VisionTS are limited. Could the authors explain this limited improvement?

5. According to Figure 6, the visual component's weight in DMMV-S is smaller than in DMMV-A. Would manually adjusting the gating value g in Equation (5) improve performance?

6. Unlike VisionTS, DMMV-A cannot perform zero-shot forecasting and requires fine-tuning. Since fine-tuning latent variable models (LVMs) may incur additional computational costs, how can one balance these increased costs against the performance gains?

**Ethical Concerns:**

["NO or VERY MINOR ethics concerns only"]

**Final Justification:**

Thank the authors for the response. The rebuttal and subsequent discussion have addressed some issues (e.g., the automatic selection of period P and the main results of DMMV-S). Several critical concerns remain unresolved:

1. The authors claim in the motivation that vision-based models exhibit strong inductive bias toward period consistency, often overshadowing the global trend. Based on this,  DMMV-S (with simple decomposition), which combines the strengths of both vision-based and time series models, should theoretically outperform the pure vision model VisionTS. Furthermore, DMMV-A (with adaptive decomposition) would be expected to surpass DMMV-S. However, the reported results indicate that DMMV-S underperforms VisionTS, which appears inconsistent with the stated motivation.

2. DMMV-A achieves remarkable results on the Weather and Electricity datasets, but the authors don’t clarify which specific designs contribute to this advantage. I believe that Figure 6, which analyzes the weights of the vision and time-series modules, is insufficient as evidence, since these weights are relatively similar across all datasets for DMMV-A.

3. Regarding the comparison with VisionTS: since DMMV-A and DMMV-S are based on modifications to VisionTS, the experimental settings should align with VisionTS’s zero-shot setup (e.g., matching the input sequence length). Such alignment would provide a fairer and more convincing demonstration of the prediction improvements brought by the DMMV models.

Given these unresolved issues, I maintain my original rating.

**Limitations:**

yes

**Quality:**

2

**Strengths And Weaknesses:**

Strengths:

1. The paper proposes a novel time series forecasting framework that combines the LVM-based forecaster and time series forecaster.

2. DMMV-A achieves good forecasting results on some datasets, indicating its potential as a new paradigm for MMV-based time series forecasting.

Weaknesses:

1. The model's forecasting performance primarily relies on VisionTS (the visual forecaster). For DMMV-A, the authors only introduce a commonly used moving average decomposition mechanism, which limits the overall novelty of the framework.

2. Compared to DMMV-A, DMMV-S involves a more complex design, yet its performance is actually worse, raising concerns about the effectiveness of the added complexity.

3. DMMV-A appears to be well-suited for datasets with periodicity. However, it remains unclear how the model would perform on non-periodic datasets—would the forecasting accuracy of DMMV degrade significantly in such cases?

---

> ### Author Rebuttal · Authors · 2025-07-31
>
> Dear Reviewer srxJ,
>
> Thank you so much for the constructive feedback. We sincerely appreciate your valuable suggestions and questions. The following are our responses.
>
> **W1: The model's forecasting performance primarily relies on VisionTS (the visual forecaster). For DMMV-A, the authors only introduce a commonly used moving average decomposition mechanism, which limits the overall novelty of the framework.**
>
> Please let us clarify a misunderstanding in this comment. DMMV-A's decomposition is based on a backcast-residual mechanism with our proposed BCmask strategy, which is different from moving average decomposition. Specifically, in Section 3.1 and 3.2, we proposed two variants of DMMV, Figure 1(a) and (b) present their structures: (a) DMMV-S (simple decomposition) adopts moving average decomposition; (b) DMMV-A (adaptive decomposition) was designed with a learnable backcast-residual based decomposition, which is adaptive. By comparing them, we find DMMV-A is better in Figure 5 (also in Table 4 in appendix (supplementary files)), thus is more recommended.
>
> We acknowledge the important role of LVMs in our method, and reflected it in the title of the paper. However, we'd like to highlight our contribution to explore multimodal views of time series by discovering and levarging a bias of the LVM-based forecaster, which could lay the groundwork for future multimodal models on time series. Also, we think the difference in the overall numbers of wins between DMMV-A and VisionTS in Table 1, i.e., 43 and 9, is remarkable, suggesting a non-trivial improvement. In particular, DMMV-A addresses VisionTS's bias toward periodic data, thus is generalizable to broader applications. As such, we hope our contributions could be fairly assessed.
>
> **W2: Compared to DMMV-A, DMMV-S involves a more complex design, yet its performance is actually worse, raising concerns about the effectiveness of the added complexity.**
>
> We guess this question is caused by the same misunderstanding in Q1. We apologize for the confusion. From Figure 1, DMMV-A's is more crafted than DMMV-S by introducing a backcast-residual decomposition, which is more adaptive. From Figure 5 (and Table 4 in appendix), this design makes DMMV-A perform better than DMMV-S. In the following, we present the average performance comparison. The full results are in Table 4. Additionally, our ablation analysis in Table 2 validates the design choices of DMMV-A.
>
> Average Performance of DMMV-A and DMMV-S across different datasets:
> | Dataset | DMMV-A(MSE) | DMMV-A(MAE) | DMMV-S(MSE) | DMMV-S(MAE) |
> | ------- | --- | --- | --- | --- |
> | ETTh1   | 0.395 | 0.414 | 0.405 | 0.426 |
> | ETTh2   | 0.337 | 0.388 | 0.339 | 0.397 |
> | ETTm1   | 0.340 | 0.371 | 0.349 | 0.382 |
> | ETTm2   | 0.256 | 0.317 | 0.254 | 0.318 |
> | illness | 1.407 | 0.771 | 1.520 | 0.813 |
> | Electricity | 0.158  | 0.248 | 0.192 | 0.296 |
> | Weather | 0.217 | 0.256 | 0.244 | 0.281 |
> | Traffic | 0.382 | 0.257 | 0.395 | 0.268 |
>
> **W3: DMMV-A appears to be well-suited for datasets with periodicity. However, it remains unclear how the model would perform on non-periodic datasets.**
>
> Quite the contrary, LVM-based forecaster alone, like VisionTS, is suited for datasets with periodicity, overlooking global trend to some extent. This is explained in our Section 3 ("An Inductive Bias"). In contrast, our method, DMMV-A, was designed to address this limitation and enhance LVM-based forecaster with multimodal views, thus is more suited for less periodic or non-periodic datasets such as **Weather** and **Illness**, as presented below.
>
> Average Performance on Non-Periodic Datasets:
> | Dataset | DMMV-A (MSE) | VisionTS (MSE) | GPT4TS (MSE) | PatchTST (MSE) |
> | --- | --- | --- | --- | --- |
> | Illness | 1.407 | 1.482 | 1.871 | 1.443 |
> | Weather | 0.217 | 0.225 | 0.227 | 0.226 |
>
>
> **Q1: This makes the model highly sensitive to the choice of segment length P; if P is not properly set, the performance may deteriorate significantly.**
>
> In fact, segment length $P$ can be set as a period by Fast Fourier Transform, releasing the burden of manual tuning. However, if $P$ is perturbed, LVM-based forecaster alone is more sensitive than our DMMV-A model since DMMV-A reduces reliance on period accuracy by adaptive decomposition. We did the following test: when the period $P$ is perturbed (i.e., varying by P/6, 2P/6, ..., P), VisionTS's performance drops more (e.g., ETTh1: 0.409 → 0.582 (42.3%)) than our model (e.g., ETTh1: 0.395 → 0.450 (13.9%)), suggesting the better robustness of our method.
>
>
> **Q2: Based on the results in Figure 5, DMMV-A outperforms DMMV-S. Does this suggest that additional design components of DMMV-S may be ineffective or unnecessary?**
>
> This question is the same as W2. The response is summarized in the reponse to W2.
>
>
> **Q3: Table 1 does not report the results for DMMV-S. Could the authors include them for completeness and comparison?**
>
> Yes. Currently, we put the results of DMMV-S in Table 4 in the appendix for keeping Table 1 succinct while highlighting salient methods. Now we are aware of the interests in DMMV-S, and we'd like to update Table 1 to include it in a revised version of the paper. Thank you for the suggestion.
>
>
> **Q4: On datasets with strong periodicity, such as Electricity and Traffic, the performance gains of DMMV-A over VisionTS are limited. Could the authors explain this limited improvement?**
>
> This question relates to W3. In fact, VisionTS is suited for datasets with strong periodicity. On such datasets, VisionTS performs well, while modeling global trend may only bring small gains. This explains why DMMV-A's gain over VisionTS is limited. In contrast, on datasets with less periodicity, such as **Weather** and **Illness**, DMMV-A achieves more notable improvements over VisionTS, validating its effectiveness in addressing VisionTS's limitation. Considering both cases, periodic or non-periodic, DMMV-A is no less than VisionTS, indicating its broader applicability.
>
>
> **Q5: According to Figure 6, the visual component's weight in DMMV-S is smaller than in DMMV-A. Would manually adjust the gating value improve performance?**
>
> In DMMV-A, $g$ is automatically learned to avoid human bias, allowing adaptive aggregation of the decomposed components. Manual tuning $g$ may be inefficient thus is not recommended.
>
> To check whether manual tuning $g$ works for DMMV-S, we set the visual weight as same as DMMV-A for DMMV-S. The following table compares the results. As can be seen, manual weights lead to performance drop. This is because the imposed manual $g$ cannot align with other parts of the model, which are aligned with the leared $g$ during model training. We'd like include this result in a revised version of the appendix.
>
> | Dataset | DMMV-S(MSE) | DMMV-S(MAE) | Manual Weight(MSE) | Manual Weight(MAE) |
> | --- | --- | --- | --- | --- |
> | ETTh1 | 0.405 | 0.426 | 0.433 | 0.452 |
> | ETTm1 | 0.349 | 0.382 | 0.376 | 0.422 |
> | illness | 1.520 | 0.813 | 1.964 | 1.313 |
> | Weather | 0.244 | 0.281 | 0.265 | 0.296 |
>
>
> **Q6: Unlike VisionTS, DMMV-A cannot perform zero-shot forecasting and requires fine-tuning. Since fine-tuning latent variable models (LVMs) may incur additional computational costs, how can one balance these increased costs against the performance gains?**
>
> Although VisionTS can be used for zero-shot forecasting, we wanted to highlight its performance is limited. The following table presents its zero-shot performance, which is even inferior to baslines DLinear, Time-LLM, PatchTST, etc. (Table 1). Thus fine-tuning is still important for VisionTS to perform optimally, like DMMV-A.
>
> On the other hand, DMMV-A allows freezing LVMs while fine-tuning other weights -- a light-weight numerical forecaster (0.2% of total weights) -- in the model, thus can be efficient. However, to reach optimal performance, fine-tuning is still recommended and it only incurs a negligible overhead compared to VisionTS, as indicated in the second table below. Consider the benefits of DMMV-A, we think the small cost overhead is paid off.
>
> VisionTS zero-shot averaged performance
> | Dataset | MSE | MAE |
> | --- | --- | --- |
> | ETTh1 | 0.4325 | 0.4178 |
> | ETTm1 | 0.4198 | 0.3945 |
> | illness | 2.312 | 1.543 |
> | Weather | 0.3195 | 0.3125 |
>
> Model size and training time
> | Model | Parameters | Training Time / Epoch |
> | --- | --- | --- |
> | DMMV-A | 112.1M | 158s |
> | VisionTS | 111.9M | 133s |

---

> > ### Comment · Reviewer_srxJ · 2025-08-05
> >
> > Thank you for your detailed rebuttal, which has addressed some of my concerns. However, I still have a few questions:
> >
> > W2: The performance of DMMV-S is notably lower than that of VisionTS, especially on the Electricity dataset. The gap is quite significant and needs to be explained.
> >
> > W2: Compared to DMMV-S, DMMV-A achieves substantial improvements on the Electricity and Weather datasets. It is important to clarify what accounts for such performance differences between the two variants.
> >
> > Q1: DMMV still appears to be sensitive to the choice of period P. Since the model transforms time series into 2D representations based on periodicity, the authors should apply a method like FFT to automatically detect the dominant periods in DMMV-A, and compare the results to validate its effectiveness.
> >
> > Q6: I believe that VisionTS demonstrates strong zero-shot performance, as evidenced by the results in the original paper. Zero-shot learning has the advantage of eliminating the need for additional training or fine-tuning, significantly reducing computational cost. Meanwhile, the reported model size and training time lack clarity—it is necessary to specify which dataset and GPU environment are used for the measurement.

---

> > > ### Author Response · Authors · 2025-08-07
> > > **Response to remaining concerns (Part 1)**
> > >
> > > Dear Reviewer srxj,
> > >
> > > Thank you for reviewing our response! We are glad that our rebuttal helps address some of your concerns. Meanwhile, please let us address your remaining concerns as follows.
> > >
> > > **1. The performance of DMMV-S is notably lower than that of VisionTS, especially on the Electricity dataset. The gap is quite significant and needs to be explained.**
> > >
> > > Thank you for pointing it out. Yes, we think the performance difference between DMMV-S and VisionTS is worth further explanation and should be added to Appendix D.1. In fact, the limitation of DMMV-S that we observed when comparing with VisionTS inspired us to develop DMMV-A.
> > >
> > > In Section 3.1 ("Remark"), we stated the limitation as: the simple moving-average (MOV) decomposition in DMMV-S will enforce its numerical forecaster and visual forecaster to **fit pre-defined components** extracted by a certain kernel size. This is not flexible and cannot fully leverage LVMs' potential. This is reflected in Figure 6, from which we can see the contribution of the LVM in DMMV-S is small, e.g., 0.35 on the Electricity dataset.
> > >
> > > This analysis indicates MOV decomposition causes DMMV-S to under-explore LVMs. In contrast, VisionTS only has an LVM backbone, thus fully, but overly, explores it. Its over-exploration of LVM leads to the bias as we discussed in Section 3 ("An Inductive Bias"). In contrast to both of them, DMMV-A does not under-explore nor over-explore LVMs and stays in the middle, i.e., adaptively learn a decomposition that fits its LVM's modeling capability. Figure 6 indicates an adaptive contribution of LVM in DMMV-A is around 0.85.
> > >
> > > Finally, we'd like to clarify the reason for keeping DMMV-S in the paper: DMMV-S is provided mostly as a baseline to indicate the need of an adaptive decomposition method like DMMV-A. Without DMMV-S, readers may be curious about the effectiveness of MOV decomposition. We are thankful to the reviewer's question and we will add this discussion to the revised draft.
> > >
> > >
> > > **2. Compared to DMMV-S, DMMV-A achieves substantial improvements on the Electricity and Weather datasets. It is important to clarify what accounts for such performance differences between the two variants.**
> > >
> > > We think our response to the above question helps explain this question. Here we'd like highlight DMMV-A's key advantage: As Figure 1(b) and Section 3.2 illustrate, DMMV-A introduces a backcast-residual mechanism that enables its LVM to determine a seasonal component from the input time series that fits its modeling capability. Through this design, the LVM can push the parts of the signal that it cannot fully interpret to the numerical forecaster for handling. In contrast, the LVM in DMMV-S is imposed with a pre-defined seasonal component. From Figure 6, DMMV-A makes LVM play a more important role while not overly dominates its numerical forecaster, providing a fit that mitigates the LVM's bias.
> > >
> > >
> > > **3. DMMV still appears to be sensitive to the choice of period P. The authors should apply a method like FFT to automatically detect the dominant periods in DMMV-A, and compare the results to validate its effectiveness.**
> > >
> > > Yes, we agree that automatically detecting periods by FFT largely helps set a proper P. In fact, to be consistent we our description in Section 3 ("Preliminaries" line 120), in our experiments, we set all periods using FFT. This can be validated from our submitted code. In file `utils.vision.py`, the functions `find_periods` and `find_periods_multi_variable` were used to set periods using FFT. Specifically, the periods for all datasets detected by FFT are summarized in the following. We will add this statement and the summary of periods to a revised version of the Section B in the appendix.
> > >
> > > | EETh1 | EETh2 | EETm1 | EETm2 | Weather | Illness | Traffic | Electricity |
> > > | --- | --- | --- | --- | --- | --- | --- | --- |
> > > | 24  |  24 |  96 |  96 | 144 | 52 | 24 | 24 |

---

> > > > ### Author Response · Authors · 2025-08-07
> > > > **Response to remaining concerns (Part 2)**
> > > >
> > > > **4. I believe that VisionTS demonstrates strong zero-shot performance, significantly reducing computational cost. Meanwhile, the reported model size and training time lack clarity, it is necessary to specify which dataset and GPU environment are used for the measurement.**
> > > >
> > > > We have carefully checked VisionTS's zero-shot experiments and its code. It is noteworthy that the zero-shot results reported in the original paper of VisionTS were obtained by using non-standarded, manually tuned look-back window sizes (e.g., 2880 for Electricity and 4032 for Weather), which can be found in Table 8 (Appendix B.1) of its paper. This leads to unfairly different samples in the test sets. We've tested VisionTS using the standard evaluation protocol (as described in Section 4 ("Datasets") in our paper) on the benchmark datasets. In our previous response to Q6, the first table summarizes VisionTS zero-shot performance, we put it in the following and compare the reproduced results and the results reported in the original paper.
> > > >
> > > > As can be seen, under standard evaluation protocol, VisionTS's zero-shot performance is even inferior to baselines such as DLinear, Time-LLM, PatchTST, etc. (please see Table 1 in our paper). Thus there is a performance-computation tradeoff. To reach a competitive performance, VisionTS still needs fine-tuning. In this case, the computational overhead of our method is small, as indicated by the second table in our previous response to Q6 (also in the second table below).
> > > >
> > > > For the model size and training time, please let us supplement their datasets and add more examples as in the second table below. All experiments were conducted on NVIDIA RTX 6000 Ada GPUs (48GB) as described in Appendix C.3.
> > > >
> > > > **Table A. VisionTS zero-shot averaged performance**
> > > > | Dataset | Reproduced (MSE) | Reproduced (MAE) | Original (MSE) | Original (MAE) |
> > > > | --- | --- | --- | --- | --- |
> > > > | ETTh1 | 0.433 | 0.418 | 0.390 | 0.414 |
> > > > | ETTm1 | 0.420 | 0.395 | 0.374 | 0.372 |
> > > > | Illness | 2.312 | 1.543 | N/A | N/A |
> > > > | Weather | 0.320 | 0.313 | 0.269 | 0.292 |
> > > >
> > > > **Table B. Model size and training time**
> > > > | Model | Parameters | Training Time / Epoch | Dataset |
> > > > | --- | --- | --- | --- |
> > > > | DMMV-A | 112.1M | 158s | EETh1 |
> > > > | VisionTS | 111.9M | 133s | EETh1 |
> > > > | DMMV-A | 112.1M | 618s | EETm1 |
> > > > | VisionTS | 111.9M | 556s | EETm1 |
> > > > | DMMV-A | 112.0M | 9.2s | Illness |
> > > > | VisionTS | 111.9M | 8.3s | Illness |
> > > > | DMMV-A | 112.1M | 1954s | Weather |
> > > > | VisionTS | 111.9M | 1764s | Weather |

---

> > > ### Author Response · Authors · 2025-08-09
> > > **Looking forward to hearing from Reviewer srxJ**
> > >
> > > Dear Reviewer srxJ,
> > >
> > > In our previous response, we tried our best to address the remaining misunderstandings. We will be very grateful if you could let us know whether our previous response helps address your remaining concerns. Also, we'd like to take the opportunity before the end of the discussion period to address any further questions. Thank you for your attention!

---

### Official Review · Reviewer_SJEM · 2025-07-03

**Clarity:** 2
**Significance:** 3
**Originality:** 3
**Rating:** 4
**Confidence:** 4

**Summary:**

This paper presents DMMV, a novel framework for multi-modal long-term time series forecasting (LTSF) that integrates latent variable models (LVMs) with an adaptive decomposition strategy. To mitigate the periodicity bias inherent in LVMs, DMMV introduces a customized backcast-residual mechanism that enables effective fusion of numerical and visual modalities. Comprehensive experiments on standard benchmarks show that DMMV consistently outperforms both single-modal models and state-of-the-art multi-modal baselines, highlighting its robustness and effectiveness.

**Questions:**

Answer above Weaknesses.

**Ethical Concerns:**

["NO or VERY MINOR ethics concerns only"]

**Limitations:**

yes

**Quality:**

3

**Strengths And Weaknesses:**

**Strengths**

S1. The research presented in this paper is relatively novel, focusing on the application of large multimodal models to time series data.

S2. This paper is technically strong and presents extensive experiments, bringing many new insights to the field.



**Weaknesses**

W1. Is the performance improvement achieved by the proposed method too incremental? A significance test may be necessary to validate its effectiveness.

W2. Are visual models sensitive to the visual representation of time series data, such as line color, pixel conditions, the proportion of the time series line in the image, line smoothness, etc.?

W3. Do multimodal models have potential applications in the spatiotemporal domain? Spatio-temporal data emphasizes the relationships between channels and may face many challenges. Can you bring more extended insights?

---

> ### Author Rebuttal · Authors · 2025-07-31
>
> Dear Reviewer SJEM,
>
> Thank you so much for the constructive feedback. We sincerely appreciate your valuable suggestions and questions. The following are our responses.
>
> **W1: Is the performance improvement achieved by the proposed method too incremental? A significance test may be necessary to validate its effectiveness.**
>
> Thank you for highlighting this concern. Yes, the challenge of improving MSE and MAE over the SOTA methods on the widely used benchmark datasets may raise doubts about the significance of improvements. To address this concern, we ran our method three times with different seeds and calculated standard deviations. In our appendix, Table 9 reports the standard deviations of DMMV on the same four datasets used for ablation and performance analysis (section 4.2, 4.3). In the following, we summarize the comparison between DMMV-A and the most competitive baseline, VisionTS.
>
> From the results, both methods show small standard deviations across multiple runs, suggesting their stability and the reliability of the results. We ran two-sample t-tests for each dataset under different forecasting horizons (i.e., 96, 192, 336, 720). DMMV-A exhibits significant improvements over VisionTS in 13 out of 16 cases using MSE (p-value<0.05) and 9 out of 16 cases using MAE (p-value<0.05), totaling a majority of the cases. Also, considering the non-trivial difference in the overall numbers of wins between DMMV-A and VisionTS in Table 1, i.e., 43 and 9, we think the improvements are not a matter of randomness. We will extend this analysis to all datasets and add the results to a revised version of the appendix.
>
> | Datasets | Predicted length | DMMV-A(MSE) | DMMV-A(MAE) | VisionTS(MSE) | VisionTS(MAE) |
> | --- | --- | --- | --- | --- | --- |
> | ETTh1 | 96 | 0.354$\pm$0.001 | 0.390$\pm$0.001 | 0.355$\pm$0.001 | 0.386$\pm$0.001 |
> |  | 192 | 0.393$\pm$0.001 | 0.405$\pm$0.001 | 0.395$\pm$0.003 | 0.407$\pm$0.000 |
> |  | 336 | 0.387$\pm$0.001 | 0.413$\pm$0.001 | 0.419$\pm$0.000 | 0.421$\pm$0.000 |
> |  | 720 | 0.447$\pm$0.002 | 0.451$\pm$0.001 | 0.458$\pm$0.002 | 0.460$\pm$0.001 |
> | ETTm1 | 96 | 0.278$\pm$0.001 | 0.329$\pm$0.000 | 0.284$\pm$0.001 | 0.332$\pm$0.001 |
> |  | 192 | 0.317$\pm$0.001 | 0.358$\pm$0.001 | 0.327$\pm$0.001 | 0.362$\pm$0.001 |
> |  | 336 | 0.351$\pm$0.001 | 0.381$\pm$0.000 | 0.354$\pm$0.001 | 0.382$\pm$0.001 |
> |  | 720 | 0.411$\pm$0.000 | 0.415$\pm$0.000 | 0.411$\pm$0.001 | 0.415$\pm$0.000 |
> | illness | 24 | 1.409$\pm$0.001 | 0.754$\pm$0.001 | 1.613$\pm$0.001 | 0.834$\pm$0.000 |
> |  | 36 | 1.291$\pm$0.002 | 0.742$\pm$0.003 | 1.316$\pm$0.002 | 0.750$\pm$0.001 |
> |  | 48 | 1.499$\pm$0.002 | 0.810$\pm$0.011 | 1.548$\pm$0.002 | 0.818$\pm$0.001 |
> |  | 60 | 1.430$\pm$0.003 | 0.774$\pm$0.001 | 1.450$\pm$0.001 | 0.783$\pm$0.001 |
> | Weather | 96 | 0.143$\pm$0.001 | 0.196$\pm$0.002 | 0.146$\pm$0.000 | 0.191$\pm$0.000 |
> |  | 192 | 0.187$\pm$0.001 | 0.245$\pm$0.003 | 0.194$\pm$0.001 | 0.238$\pm$0.001 |
> |  | 336 | 0.237$\pm$0.001 | 0.272$\pm$0.003 | 0.243$\pm$0.000 | 0.275$\pm$0.000 |
> |  | 720 | 0.300$\pm$0.002 | 0.318$\pm$0.003 | 0.318$\pm$0.004 | 0.328$\pm$0.001 |
>
>
> **W2: Are visual models sensitive to the visual representation of time series data, such as line color, pixel conditions, the proportion of the time series line in the image, line smoothness, etc.?**
>
> This is an interesting question. If time series is imaged by line plots, we saw a survey paper [*] discussed it (in its Section 3.1). Its conclusion is that there is no consensus on whether graphical components, such as lines, colors, legend, grids and tick labels, could provide extra benefits in any task.
>
> In our method, since we didn't use line plots, our method can be think as not being influenced by graphical components. Section 3 (Preliminaries) describes our imaging method. Time series is segmented by period. The segments are stacked to form a matrix, which is resized to a 224x224x3 gray-scale image. The image is gray because the 3 channels are the same. This is intuitive because we don't have extra information to be encoded by color. The most informative pattern is the texture of the image, which is determined by time series values. As such, we think our method will not introduce sensitivity to graphical components to vision models.
>
> [*] J. Ni et al. "Harnessing Vision Models for Time Series Analysis: A Survey". In IJCAI, 2025.
>
> **W3: Do multimodal models have potential applications in the spatiotemporal domain?**
>
> We are thankful to this great comment. We believe spatiotemporal data modeling is a notable direction to us for its wide applications in domains such as traffic, climate and neuroscience. One key challenge is modeling spatial dependency across channels together with their dynamics. In our humble opinion, our method can be extended to spatiotemporal data by introducing a spatial encoder, which could be a Transformer-like architecture that learns the attention among channels through intermediate embeddings. However, a bottleneck in computational cost may appear when there are hundreds or thousands of channels, which is worth further research. The key advantage of using multimodal view models, like ours, includes their capabilities in capturing complementary patterns across views and leveraging cross-modal knowledge transfer from LVMs, especially when task-specific data are scarce. We'd like to add this discussion to the revised appendix.

---

> > ### Comment · Reviewer_SJEM · 2025-08-05
> >
> > Thank you for the rebuttal. I have no further questions.

---

> > > ### Author Response · Authors · 2025-08-05
> > > **Thank you for reviewing our response!**
> > >
> > > Dear Reviewer SJEM,
> > >
> > > Thank you for reviewing our response! We are glad that our response helps address your concerns. We would also appreciate it if you could let us know what might be missing so that we can take the opportunity to clarify it and improve our work to meet your standards for a more positive rating. Thank you!

---

### Note · Authors · 2025-08-12

Dear Reviewers and Area Chairs,

We sincerely thank all the reviewers for the constructive comments on our work. We appreciate the reviewers' recognition of the merits of our work, including the finding on the bias of the SOTA LVM-based forecaster, the effective solution to address the limitation by a backcast-residual based adaptive decomposition framework, the experiments with 14 SOTA baselines categorized by vision, language, multimodal, and numerical methods, and the potential in inspiring future research on LVM-based forecasters.

In our author response, we've tried our best to reply to all comments. We appreciate that Reviewers SJEM, KH6N and Ymur acknowledged our response without posting further significant concerns. While we have not yet received the final comment from Reviewer srxJ on our latest response, we hope our response may have addressed all the major misunderstandings, particularly the questions that confuse the proposed methods DMMV-S and DMMV-A and that pertain to their differences.

Finally, we will follow the reviewers' suggestions by including in the revised draft and appendix the supplementary results we presented in our author response, along with the key insights we obtained when addressing the questions.

---

### Decision · Program_Chairs · 2025-09-17

**Decision:**

Accept (poster)

**Comment:**

Most reviews are positive, specifically from BA, BA, BA, and BR. The majority of reviewers recognize the merits of the proposed work, including the findings on the bias of the SOTA LVM-based forecaster, the effective solution to address limitations through a backcast-residual based adaptive decomposition framework, the comprehensive experiments involving 14 SOTA baselines categorized by vision, language, multimodal, and numerical methods, and the potential to inspire future research on LVM-based forecasters.
The authors have made substantial efforts to address the concerns raised by the reviewers, and most of these concerns have been resolved. Reviewers SJEM, KH6N, and Ymur have no further concerns. The authors have also provided detailed responses to the questions posed by Reviewer srxJ, and I believe the remaining questions have been adequately addressed.
Therefore, I recommend the acceptance of this submission. Additionally, I expect that the authors will incorporate the suggested modifications from the rebuttal phase into the final version.